



# A Bayesian Approach to Infer Nitrogen Loading Rates from Crop and Landuse Types Surrounding Private Wells in the Central Valley, California

Katherine M. Ransom[1], Andrew M. Bell[2], Quinn E. Barber[3], George Kourakos[1], and Thomas Harter[1]

[1]Department of Land, Air, and Water Resources, University of California, Davis, USA.
[2]Center for Watershed Sciences, University of California, Davis, USA.
[3]Department of Renewable Resources, University of Alberta, Edmonton, Canada.

*Correspondence to:* Thomas Harter (thharter@ucdavis.edu)

**Abstract.** Nitrate contamination of alluvial aquifers in agricultural areas is a typical and major problem around the world. Nitrogen applied to crops, in the form of synthetic fertilizers or manure, in excess of plant uptake, largely leaches to groundwater in the form of nitrate, which is stable and highly mobile in oxygen-rich groundwaters. Increased awareness of the impact that excess nitrogen has had on groundwater and major health concerns about nitrate are prompting new regulations for farmers,

e.g., in Europe and California, USA. This study is focused in the Central Valley, California, USA, an intensively farmed region with high agricultural crop diversity. Though nitrogen loading rates for several crop and landuse types in the Central Valley have been estimated or measured in a handful of studies, nitrogen loading rates for specific crop or landuse types and their impact to groundwater quality remain largely unknown. Knowledge of crop or other landuse specific groundwater nitrate impact may aid future regulatory decisions. Nitrogen loading rates for specific crop or landuse types are expected to vary depending on

individual landuse practices; and interactions with hydrogeologic parameters that may promote or inhibit nitrate leaching. In this study, we developed a novel Bayesian regression model that allowed us to estimate crop or other landuse-specific groundwater nitrogen loading rate probability distributions from surveys of private wells, each of which is likely impacted by more than one landuse. We used recent nitrate measurements from 2149 wells in the Central Valley. We estimated nitrogen loading rate distributions for 15 crop and landuse groups. These were shown to compare favorably with prior mass-balance estimates

of loading rates based on agronomic estimates of nitrogen loading.

## 1   Introduction

Nitrate contamination of groundwater is a common problem in agricultural regions across the globe that has also gained increased regulatory attention in recent years. The European Union Nitrates directive, which strives to protect groundwater quality across Europe, reported that as of 2010 all 27 member states had developed action programs to cut nitrogen pollution.

These action programs include monitoring networks, nitrogen application limits, and new technologies for nutrient processing (European Commission, 2010). In California, USA, the Irrigated Lands Regulatory Program (ILRP) was created in 2003 to regulate agricultural water discharge to surface water. In 2012, the ILRP was updated to issue permits for discharge to





groundwater and all commercial agriculture is now regulated under the program (Irrigated Lands Regulatory Program, FAQ, February 2016, http://www.waterboards.ca.gov/centralvalley/water_issues/irrigated_lands/ilrp_faq.pdf).

Several studies have documented the presence of nitrate contamination in shallow groundwater of the Central Valley aquifer system (Burow et al., 2012, 1998b; Boyle et al., 2012; Lockhart et al., 2013; California State Water Resources Control Board, 2010), where this study is focused. Many people in the Central Valley rely on shallow private wells for domestic use. Studies estimate the number of private wells in the Central Valley to be on the order of 100,000 to 150,000 (Viers et al., 2012; Johnson and Belitz, 2015). The federal drinking water standard to protect against methemoglobinemia (blue-baby syndrome) is 10 mg/L $NO_3$-N. Drinking water with nitrate concentrations above background levels of near 1 mg/L $NO_3$-N, but below the drinking water standard, has also been linked to an increased risk of ovarian cancer (Inoue-Choi et al., 2015), thyroid cancer (Ward et al., 2010), bladder cancer (Weyer et al., 2001), and non-Hodgkin's lymphoma (Ward et al., 1996).

Nitrate contamination of groundwater may originate from several sources including synthetic fertilizer, manure, septic systems, and leaky sewer lines. In the Central Valley, California, the major contributors to nitrate contamination are fertilizers and manure applied to crops (Rosenstock et al., 2013; Harter and Lund, 2012). Knowledge of highest-risk crops can aid future regulatory efforts and help in defining priority areas on a county or smaller scale. Previous field-based, often plot-scale research, conducted in California, has measured the amount of nitrogen leached in kg ha$^{-1}$ yr$^{-1}$ from several different crop types including, grains, vegetables and berries, citrus, nuts, and field crops (Devitt et al., 1976; Embleton et al., 1979; Letey et al., 1977; Pratt et al., 1972; Pratt and Adriano, 1973; Adriano et al., 1972; Allaire-Leung et al., 2001; Liang et al., 2014). However, these studies were largely conducted in the 1970s and 1980s, with little research conducted since then, and are subject to high variability due to various measurement methods (Viers et al., 2012). For example, the historical measurements of nitrogen loading for vegetable and berry crops range from about 20 kg N ha$^{-1}$ yr$^{-1}$ to over 900 kg ha$^{-1}$ yr$^{-1}$. Two studies estimate nitrogen loading rates from several crop or landuse groups in the Central Valley based on a field-scale nitrogen mass balance approach (Viers et al., 2012; Rosenstock et al., 2013). Loading rates for Central Valley dairy corrals, lagoons, and forage crops receiving manure applications have been estimated based on groundwater monitoring wells located on dairies (Harter et al., 2002; van der Schans et al., 2009). What is lacking in the literature is a regional scale assessment of nitrogen loading to groundwater based on measured groundwater nitrate data.

The main objective of this paper was to investigate what information may be obtained from existing groundwater quality data that could reveal nitrate loading rates from the large diversity of crop types within the Central Valley. More than 250 crops are grown within the Central Valley alone. With a statistical framework, we estimated crop and other landuse-specific nitrogen loading rates (kg N ha$^{-1}$ yr$^{-1}$) to groundwater from well nitrate data and historic crop and landuse information around each well. Nitrate loading concentrations vary within specific crop or landuse types due to differences in farming practices and interactions between hydrogeologic parameters and those practices (farm to farm variation). Loading rates also vary across an individual farm due to small-scale hydrologic heterogeneity (within farm variation). We therefore expect a range of loading rates to apply to any given crop or landuse type. Furthermore, significant uncertainty exists about the source area and the resulting landuses that contributed to a well's nitrate concentration.




Bayesian statistical models have specific benefits when it comes to dealing with uncertainty and complex interactions in groundwater systems: they can incorporate prior knowledge, take into account uncertainty, and allow for variability in predictions. We develop a generalized linear model using Bayesian methods and estimate a probability distribution of nitrate leaching for crop or landuse types. In contrast to prior studies, estimates here were not based on agronomic data, but on a comprehensive groundwater quality dataset and a geospatial analysis of crops or other landuses surrounding each well. We compiled two large datasets: a database of nitrate measurements from private wells distributed throughout the Central Valley for use in the model and a crop and landuse analysis in the most likely region to have affected an individual well's nitrate concentration.

## 2    Project area

The Central Valley (CV), California is a large asymmetric, alluvial basin, with the trough axis trending slightly north-west to south-east. The CV is 400 miles long (extending from Red Bluff to Bakersfield, CA), an average of 50 miles wide, and has an area of approximately 20,000 square miles. The boundaries of the CV are the Cascade Range to the north, the Coast Ranges to the west, the Sierra Nevada mountains to the east, and the Tehachapi Mountains to the south. The CV consists of two separate valleys, divided at the Sacramento-San Joaquin Delta: the Sacramento Valley to the north (northern one-third) and the San Joaquin Valley to the south (southern two-thirds). The CV is filled with six to ten miles of marine and continental deposits. Surface geomorphology consists of overflow lands and lake bottoms, river floodplains and channels, low alluvial plains and fans, and dissected uplands. Post-Eocene continental deposits consist of fine to coarse sediments and compose the major aquifer in the CV (Page, 1986). Spring 2011 depth to groundwater ranged from 10 feet below ground surface (bgs) in the northern section of the CV to 670 feet bgs in the southern portion of the CV (DWR, 2011).

The Central Valley is a highly productive agricultural region with approximately 7 million of California's nearly 9 million acres of irrigated farmland (California Department of Water Resources, Agricultural Land and Water Use Estimates, 2010, available at http://www.water.ca.gov/landwateruse/anlwuest.cfm). Major crops grown in the CV are corn, grain and hay, oranges, almonds, peaches and nectarines, cotton, and wine and table grapes. A combination of surface water and groundwater is used for irrigation. In addition, over 80% of California's 1.8 million adult cows live on dairies in the CV. Total human population for the 19 counties associated with the CV was approximately 7 million in 2014. Major CV cities with over 200,000 residences are Sacramento, Fresno, Bakersfield, Stockton, and Modesto (United States Census Bureau, 2010 Census of Population, State and County QuickFacts, quickfacts.census.gov/qfd/states/06000.html). Residences located in rural unincorporated areas, many of them clustered in semi-urban belts around smaller towns and major cities, rely on shallow private wells for drinking and household purposes. Private wells are not regulated in California and it is difficult to know how many may be contaminated by nitrate: a large portion of the CV has been estimated to have shallow groundwater nitrate concentrations above the drinking water standard (Nolan et al., 2014; Lockhart et al., 2013).





## 3 Methods

### 3.1 Well and landuse data

A database of nitrate measurements from private wells located in California was compiled from several data sources. The California Ambient Spatio-Temporal Information on Nitrate in Groundwater (CASTING) includes nitrate measurements from

private supply, public supply, irrigation, and monitoring wells (Boyle et al., 2012). We selected all well measurements from the CASTINGS database from supply wells (not monitoring wells) designated as private. We selected private well samples because they are typically more shallow than public supply or irrigation wells and are not purposefully located near sources of contamination as are monitoring wells. Private well samples within the CASTING database initiated from several sources including the Central Valley Regional Water Quality Control Board (CVRWQCB) Fresno Office dairy domestic wells mon-

itoring data (sampled for nitrate as a part of the Dairy General Order regulations for dairy facilities in the Central Valley), the California Department of Pesticide Regulation (CDPR), Fresno County, the United States Geological Survey (USGS), Tulare County Environmental Health (TCEH), and the State Water Resources Control Board (SWRCB) Groundwater Ambient Monitoring Assessment (GAMA) Domestic Wells Project in Tulare County. We expanded the original database, which was geographically limited to the southern Central Valley, to include the data from the same data sources for the entire Central

Valley. Also, additional private well samples from the following data sources were added to the CASTINGS database:

- a set of private wells previously sampled for nitrate as a part of the "Proposition 50 Long Term Risk of Groundwater and Drinking Water Degradation from Dairies and Other Nonpoint Sources in the San Joaquin Valley", funded by the State Water Resources Control Board (SWRCB) (Lockhart et al., 2013) (200 wells total, sampled between 2010-2011),

- additional SWRCB GAMA private wells for Tehama, El Dorado, and Yuba county project areas (GAMA Domestic Well

Project, http://www.waterboards.ca.gov/gama/domestic_well.shtml) downloaded from the GeoTracker GAMA online database (http://www.waterboards.ca.gov/gama/geotracker_gama.shtml), and

- CVRWQCB Rancho Cordova Office dairy domestic wells monitoring data provided by the CVRWQCB office.

The well database was filtered for records collected between the years 2000 to 2015. Locations with data collected in multiple years were assigned the median nitrate value of all the recorded measurements. Prior to median aggregation, non-detect nitrate

values were replaced with the detection limit and zero values were replaced with the most common detection limit of 2.21 mg/L $NO_3$-$NO_3$. All nitrate measurements were then converted to $NO_3$-N by dividing by the mass ratio of 4.4268. When geographic coordinates (latitude and longitude) of the private wells in the dairy monitoring program were not available, the wells were located using the dairies street address and placed at the centroid of a dairy's land parcels. The methods for locating the wells varied for each of the other data sources including geographic coordinates, geocoded addresses, offsets by a ransom

small distance, United States Public Land Survey System (PLSS) section, and Assessor's Parcel Number (APN) (Table 1). Due to the well location methods, many wells had overlapping locations. Where multiple wells were geolocated to a single





location, a single well was chosen at random to represent that location. Wells outside of the alluvial aquifer system boundary were excluded from the analysis. The final nitrate database had a total of 2149 wells.

Intrinsic aquifer properties were evaluated as an indicator for additional risk for or protection from nitrogen contamination. Here we choose a simple binary indicator: California Department of Pesticide Regulations (CDPR) Groundwater Protection Ar-

eas (GWPAs) are one square-mile sections that are vulnerable to the leaching of pesticides and are defined by the following criteria: previous detections of pesticides in that section, contains coarse soils and a depth to groundwater less than 70 feet, or contains runoff-prone soils and depth to groundwater less than 70 feet (CDPR, http://www.cdpr.ca.gov/docs/emon/grndwtr/vasmnt.htm). These zones are either vulnerable to contamination due to non-point source leaching of irrigation water (leaching GWPAs) or direct flow paths through hardpan soils (ditches, dry wells, poorly sealed wells, runoff GWPAs)

(CDPR, http://www.cdpr.ca.gov/docs/emon/grndwtr/contproc.htm). The properties that lead to vulnerabilities from pesticide contamination also increase the possibility of nitrate contamination. Wells located within a GWPA zone were attributed as being an indicator for increased risk of nitrate contamination.

Landuse surrounding wells was analyzed using the California Augmented Multisource Landcover (CAML) raster for the year 1990 (Hollander, 2013). CAML has been developed from various datasources for five periods of five years each centered

on 1945, 1960, 1975, 1990, and 2005. The 1990 CAML time period was selected because is the earliest historical landuse map for which landuse data were digitally mapped rather than estimated by back simulation. Older maps (pre-1990) are simulated based on Agricultural Commissioner reports (Viers et al., 2012). The earlier rather than current landuse coverage was chosen to account for some of the time difference between nitrate leached from landuse practices and the time of groundwater sampling (nitrate travel time) (Ransom et al., 2016). The nearly 60 agricultural and many non-agricultural landuse categories

were aggregated into the following land type groups: Water & Natural, Citrus & Subtropical, Tree Fruit, Nuts, Cotton, Field crops, Forage crops, Rice, Alfalfa & Pasture, Confined Animal Feeding Operation (CAFO), Vegetables & Berries, Peri-Urban, Grapes (including wine and table), and Urban. The Forage Crop group was further separated into fields likely receiving liquid manure irrigation and fields not likely to receive liquid manure based on proximity to CAFO landuse (within 1 mile of dairy corrals, lagoons, or facility barns). This analysis does not take into consideration dry manure that may be exported off dairies

and applied to crops. Our final study design had a total of 15 landuse and crop groups (hereby referred to as scenario 1). Alternatively, we also analyzed a scenario where CAFO landuse is grouped with Manured Forage Crops (14 landuse and crop groups, hereby referred to as scenario 2). The approach presented here can easily be modified to other landuse or management practice categorizations.

Groundwater flow direction at each well is highly variable due to local pumping from numerous surrounding wells and is

impossible to determine without installing observation wells at each well site. Previous studies have used a circular "buffer" zone around each well as an approximation of the well source area (Burow et al., 1998a; McLay et al., 2001; Kolpin, 1996; Lockhart et al., 2013). Circular well buffers have been shown to be reasonable approximations of well source area when the actual contributing source area is unknown (Johnson and Belitz, 2009). Landuse amounts (in $m^2$) were quantified within a circular well "buffer" of radius 2.4 km surrounding each well and then converted to a percent of buffer area. The 2.4 km buffer

radius was determined by assuming a fixed vertical groundwater recharge rate of 0.30 m year$^{-1}$, effective horizontal hydraulic



conductivity of 30.5 m/day, and hydraulic gradient of 0.001 and then by the use of Darcy's Law to find specific discharge. Assuming that a private well is very low flow, is screened along its entire depth below the water table, and only intercepts passing water, source area length can be calculated by multiplying well depth by the ratio of specific discharge to groundwater recharge (Harter et al., 2002; Horn and Harter, 2009) (calculation details available in Lockhart et al., 2013).

Groundwater recharge rates can also be variable and were not assumed to be fixed for the purposes of estimating the nitrogen loading rates. Probable vertical groundwater recharge rates (m year[-1]) were estimated based on results of the Central Valley Hydrologic Model (CVHM) (Faunt, 2009). The CVHM is a highly detailed computer model that simulated monthly surface and groundwater flow components throughout the CV for a period of over 40 years. For the purposes of the CVHM, the CV was spatially divided into 20,000 model grid cells (1.6 km$^2$ each) and 10 depth layers (to a depth below ground surface

of approximately 550 m). The CVHM further divides the CV into nine textural regions based on estimated aquifer texture. Temporal discretization of the CVHM consisted of 12 monthly stress periods beginning in October 1961 and ending October 2003. For the purposes of our study, flow (m$^3$ day$^{-1}$) below the bottom of the upper model layer (50 ft deep across the majority of the CV) for each monthly stress period occuring in the 1990 decade, was averaged by textural region: we calculated total yearly average flow below the upper CVHM model layer for each CVHM texture region and each year in the 1990 decade

as an estimate of vertical groundwater recharge per year. We used CVHM 1990 model outputs in order to remain consistent with the selected landuse time period. This gave 90 estimates of probable groundwater recharge rates (for each of nine textural zones and each of ten years).

## 3.2   Statistical methods

Bayesian analysis was chosen here as it allows for the estimation of the entire probability distribution of landuse-specific nitrate

leaching concentrations rather than a deterministic value only. The variability in loading rates arise from variation in farming practices (farm to farm variation) or in hydrogeologic and pedologic parameters (within farm variation). Uncertainty about the landuse or combination of landuses contributing to a well sample, despite the availability of regional groundwater flow maps (DWR, 2011) and models (Faunt, 2009) arises from hydrological heterogeneity and largely unknown spatio-temporal variability in large scale groundwater pumping near sampled wells. These factors can greatly alter groundwater flow and thus

the source area of a well, often seasonally.

Probability distributions for loading rates for the landuse or crop groups described in Section 3.1 were estimated with an exponential distribution generalized linear model using Bayesian methods. The basic model equation is given by:

$$C_i = \frac{1}{\lambda_i} = \sum_{j=1}^{n} \beta_j A_{ij} * (0.1/r) * (I_i + (1 - I_i) * k) \tag{1}$$

where $C_i$ is the expected nitrate value for well $i$ ( mg/L NO$_3$-N), $\lambda_i$ is the parameter of the exponential distribution, n is the

number of landuse categories considered, $\beta_j$ is the unknown nitrogen loading rate from landuse $j$ (in kg N ha$^{-1}$ yr$^{-1}$), $A_{ij}$ is the percent of well buffer $i$ that is landuse $j$, 0.1 is a conversion factor to convert units of mass to units of concentration based on vertical groundwater recharge rate (with units [m*(mg/L)/(kg ha$^{-1}$)])(Pratt et al., 1972), $r$ is the vertical groundwater recharge





rate (in m yr$^{-1}$) , $I_i$ is an indicator variable representing whether or not well $i$ is located within a GWPA (0 for outside and 1 for within), and $k$ is a groundwater protection parameter representing a mean decrease in nitrate values applied only to wells outside GWPA zones. We assume nitrate is the dominant form of nitrogen within and persisting in the saturated zone (Liao et al., 2012). The percent of each landuse surrounding each well was calculated as described above.

Bayesian analysis requires the assumption of an initial probability distribution for each parameter to be estimated and this represents the current estimate or knowledge of the parameter. The initial probability distribution assigned to each parameter to be estimated is known as a prior probability distribution or "prior". Priors are updated in the modeling process and the final estimate is known as a "posterior" probability distribution.

Student t-distribution priors, representing an initial approximation of the potential nitrogen loding rate, were assumed for
$\beta_j$. For an un-biased assessment, each of the landuse or crop groups were given the same t-distribution prior which represents the potential and unknown loading rate. The location parameter of the common t-distribution prior was set to equal the median measured nitrate value of approximately 5 mg/L (see Results). Reasonable, non-informative degrees of freedom and scale are 1 and 25, respectively. The choice of t-distribution parameters reflect a heavy-tailed/high variance distribution which gives the model flexibility to move the final predicted loading rates (posterior distributions) away from the prior distribution based on
evidence observed in the measured nitrate values and surrounding landuse. The t-distribution priors were truncated at zero in order to ensure that the estimated concentrations cannot be negative.

A non-informative Student-t prior probability distribution was used as an initial approximation of the GWPA factor $k$. Appropriate location, scale, and degrees of freedom were 0.5,1, and 1, respectively (see Results section). A log-normal distribution was used for the prior probability distribution of potential recharge rates with location and scale of -2 and 0.6, respectively (see
Results section). The prior probability distribution for the GWPA factor was truncated at zero to eliminate any potential for negative values. The recharge rate, $r$ was assumed to be positive.

Markov Chain Monte Carlo (MCMC) methods were used to infer the marginal posterior distributions of nitrogen loading rates for each landuse. MCMC was performed using the Gibbs sampler JAGS (Martyn Plummer, 2003, JAGS: A program for analysis of Bayesian graphical models using Gibbs sampling, version 4.2.0). JAGS was run from within the statistical
computing program, R (R Core Team, 2016, R: A language and environment for statistical computing, R Foundation for Statistical Computing, Vienna, Austria, http://www.R-project.org/), using the package rjags (Martyn Plummer, 2016, rjags: Bayesian Graphical Models using MCMC, R package version 4-6, http://CRAN.R-project.org/package=rjags). Two chains were run and the first 100,000 realizations of each chain were discarded as the burn-in period. After burn-in, each chain was sampled 200,000 times with a thinning interval of 400 to reduce autocorrelation. Traceplots were visually inspected to confirm
convergence and proper mixing of chains, after which the realizations from each chain were combined for a total of 1,000 realizations per parameter. Final model run time was approximately 12 hours on a Intel Core I-5 4670 processor with 16 GB of 1600 MHz DDR3 RAM.

We calculated a standardized Pearson goodness-of-fit statistic (McCullagh and Nelder, 1989) for each scenario. The goodness-of-fit statistic used the average of the raw residuals between the measured nitrate value for each well and the estimated value
(realization) divided by the standard deviation of all of the realizations for each well (500). The sum of the squared average





raw standardized residuals was then divided by the total number of wells minus the number of parameters in the model (17 for scenario 1 and 16 for scenario 2) to give the goodness-of-fit.

# 4    Results

The wells in the database compiled for this study had a minimum nitrate concentration of non-detect, median of 4.95, and

maximum of 131.0 mg/L (1). Median nitrate value for wells located within GWPAs was more than twice the median nitrate value for wells located outside GWPAs (8.22 mg/L $NO_3$-N versus 3.92 mg/L). The non-parametric Kruskal-Wallis test for independence between the two groups of well's nitrate values was significant at the 95% confidence level. The fact that GWPA zone wells had higher nitrate than non-GWPA zone wells justifies the use of a GWPA related groundwater protection term in the model (Equation 1). Median nitrate value was calculated for runoff versus leaching GWPAs and the values were within 1

mg/L, therefore runoff and leaching GWPA zones were grouped for the context of this study (Section 3.1). The model estimated posterior distribution for the non-GWPA protection factor, $k$, was fairly narrow, with a 95% credibility interval (CI) of between 0.698 to 0.846 and a median of 0.773 (Figure 2).

Median depth to top and bottom of well screen for all wells with depth information available (915 wells) was 40.5 m (133 ft) and 64 m (210 ft), respectively. Minimum and maximum depth to top of well screen was 2 m (6 ft) and 184.5 m (602 ft),

respectively and minimum and maximum depth to bottom of well screen was 6 m (20 ft) and 274.5 m (900 ft).

Estimated groundwater recharge from the CVHM model used as the basis for the recharge rate ($r$, Equation 1) parameter was between 0.016 and 0.530, with a median of 0.146 (m year$^{-1}$) (Figure 3). Our Bayesian model estimated (updated, posterior) recharge rate was slightly greater (posterior 95% credibility interval of 0.161 to 0.490 and median of 0.281 m year$^{-1}$), but still within the range of the CVHM model estimates (Figure 3).

Nitrate concentrations for each well were compared to the percent of each landuse or crop group within well buffers for scenario 1 with a locally weighted scatterplot smoothing (lowess) line for each plot (Figure 4). A lowess line is a smoothed regression line that represents many locally weighted polynomial fits to the data by weighted least squares (Cleveland, 1979). As Citrus & Subtropical crops, Manured Forage crops, and CAFO landuse proportions increased, nitrate in well samples appeared to increase (these groups should have greater predicted nitrogen loading rates).

The Pearson goodness-of-fit statistic, a summary measure of squared deviations between observations and their estimated values, has a value of near 1 for good-fitting models (McCullagh and Nelder, 1989). The standardized Pearson goodness-of-fit statistics were 1.16 and 1.17, respectively for scenario 1 and 2 and therefore the model was understood to fit the data fairly well.

Estimated potential nitrogen loading rates across all crop and landuse groups for both scenarios ranged from between negli-

gibly small to nearly 600 kg N ha$^{-1}$ yr$^{-1}$ (Figure 5). The scenario 1 CAFO group and the scenario 2 Manured Forage & CAFO group had the greatest estimated nitrogen loading rates, while Alfalfa & Pasture and Water & Natural for both scenario 1 and 2 were the lowest (Figure 5 and Table 2). The scenario 1 CAFO group also had the greatest range of estimates, reflecting the highest degree of uncertainty or spatial variability. Results for both scenario 1 and 2 were fairly consistent, with the exception





of the scenario 1 CAFO and scenario 2 Manured Forage & CAFO groups. When Manured Forage, which occupies relatively large areas, was grouped with nitrogen intensive, but small area CAFO landuse in scenario 2, the estimate for Manured Forage & CAFO was over three times lower than the estimated nitrogen loading for CAFO alone (Tables 2 3), but it was about two times higher than for Manured Forage alone in scenario 1. The effect of merging the two groups was much less drastic for the other crops and landuses; several estimated loading rates increased slightly, such as for Vegetable & Berry crops, while others decreased slightly (Field crops), or remained approximately the same.

Scenario 1 results were compared to historical direct measurements of nitrogen loading to groundwater from selected crops in California (Figure 6). Historical measurements of nitrogen loading from previous studies were made with the use of soil samples, anion exchange resin bags, suction lysimeters, or tile drain samples (Devitt et al., 1976; Embleton et al., 1979; Letey et al., 1977; Pratt et al., 1972; Pratt and Adriano, 1973; Adriano et al., 1972; Allaire-Leung et al., 2001; Liang et al., 2014). Results were generally consistent with historical direct measurements of nitrogen loading; each crop group with historical direct measurements had multiple measurements overlap the scenario 1 95% credibility interval of nitrogen loading (all three direct measurements for rice overlap the scenario 1 estimates for Rice). Though, historical direct measurements encompass a wider range of values and extend to greater values than our scenario 1 estimates, especially for the Vegetables & Berries crop group.

Scenario 1 results were also compared to Central Valley-wide mass-balance estimates of field-scale nitrogen loading for selected crop or landuse groups from the Groundwater Nitrogen Loading Model (GNLM) (Viers et al., 2012; Rosenstock et al., 2013) (Figure 7). GNLM and scenario 1 results are fairly consistent for Citrus & Subtropical, Vegetables & Berries, Field crops, Grapes, and the Water & Natural group. However, GNLM estimates are greater for Manured Forage crops, Nuts, Cotton, Tree Fruit, and Rice. The 95% credibility intervals (CIs) for GNLM estimates of nitrogen loading from Manured Forage crops and Cotton were wide and covered a much greater range of values than the scenario 1 estimates from this study for the same crop groups.

Other single value mass balance based estimates of nitrogen loading are within our model estimated CIs, while some are not. GNLM assigned 20 kg N ha$^{-1}$ yr$^{-1}$ to urban landuse, within our model estimated CI for Urban nitrogen loading in both scenario 1 and 2. Our model estimated CIs for Peri-Urban areas was near the range estimated by Viers et al. (2012) of 10-50 kg N ha$^{-1}$ yr$^{-1}$ for nitrogen loading from septic systems based on average human nitrogen excretion rates and estimated septic system density for their study area. The GNLM study assigned 183 kg N ha$^{-1}$ yr$^{-1}$ for dairy corrals, and 1045 kg N ha$^{-1}$ yr$^{-1}$ for dairy lagoons, both landuse groups included in our CAFO category. The GNLM value assigned for corrals is within our model estimated CI for CAFO (scenario 1), however, the GNLM value assigned to lagoons is nearly twice a high as the upper bound of the scenario 1 estimated loading rate for CAFO. Other previous point estimates from van der Schans et al. (2009) for dairy corrals, lagoons, and manured forage crops (872, 807, and 486 kg N ha$^{-1}$ yr$^{-1}$, respectively) are greater than our model estimates for CAFO or Manured Forage crops. Another mass balance estimate for manured forage fields of 280 kg N ha$^{-1}$ yr$^{-1}$ (Harter et al., 2002) are also greater than our model estimates for Manured Forage crops.

CAML landuse groups for scenario 1 are plotted side by side with corresponding Bayesian model estimates median nitrogen loading rate in order to spatially represent relative risk to groundwater from nitrate contamination (Figure 8). Low estimated





nitrogen loading is concentrated in the northern CV, where Water & Natural landuse and Rice crops are dominant, along the middle eastern and southern eastern and western edge where Water & Natural landuse is dominant, or scattered along the central axis of the CV following the pattern of Alfalfa & Pasture crops. The greatest nitrogen loading is scattered randomly throughout the central CV from north to south (a direct representation of CAFO locations), or echoes the pattern of Citrus &

Subtropical crops along the eastern edge of the southern CV in Tulare and Kern counties.

## 5    Discussion

The Bayesian estimation model provides an independent evaluation of nitrogen loading rates to groundwater from various landuses. Unlike the comparison data (Figure 6), the model here represents an inverse model estimate of nitrogen loading to groundwater using information from a large number of groundwater production wells. Using the relative proportion of landuse

groups within the source area and estimating recharge rates, groundwater concentrations are transformed to nitrogen loading rate distributions for the landuse groups.

In general, model estimates were consistent with previous independent estimates and measurements of nitrogen loading, thus demonstrating the general usefulness and accuracy of the proposed Bayesian loading model. For example, the loading rates within the credibility interval (CI) of the Alfalfa & Pasture and Water & Natural groups had the lowest overall values (Figure

5). Low estimated nitrogen loading rates are expected for both landuse categories as fertilizers and manure are not typically applied to these areas (alfalfa is a legume, which has the ability to fix atmospheric nitrogen). In addition, there was no apparent correlation between an increase in nitrate concentration in wells and increasing proportions of these two landuse groups within well buffers (Figure 4).

The CAFO landuse group and the Citrus & Subtropical crop group, however, both had positive apparent correlation between

nitrate concentration in wells and proportion of well buffers as either CAFO or Citrus & Subtropical (Figure 4). Among all the landuse or crop groups, CAFO had the greatest estimated median loading rate (a 272.82 kg N ha$^{-1}$ yr$^{-1}$ over four times the next greatest median rate of 64.60 for Citrus & Subtropical, Table A2) and also the widest 95% CI. The CI did not overlap any of the other estimated 95% CIs for the remaining landuse or crop groups (Figure 5 and Table 2).

The high loading rates estimated by our model for Citrus & Subtropical crops are not surprising, given the potential for direct

contamination pathways induced by farming practices thought to be common in the area: the majority of Citrus & Subtropical crops within the CV are located along the eastern edge of the valley floor in Tulare and Fresno counties (Figure 8). These crops are predominately grown within "runoff" designated GWPA zones on soils that contain a shallow hardpan layer (Troiano et al., 2014) and where dry wells used for surface drainage are common (DeMartinis and Royce, 1990). In addition, the water table in these same regions is relatively shallow (20-30 ft bgs) (DWR, 2011). Infiltration of agricultural surface runoff through dry

wells and/or a shallow depth to water could lead to the greater nitrogen loading predicted by our model for this crop group than for other crops/landuses. We therefore consider our estimates for Citrus & Subtropical crops robust.

Peri-Urban areas have a greater predicted CI when compared to Urban (Figure 5) and our estimates agreed with previous mass balance based estimates (See Section 4). Peri-Urban areas are defined as rural homesteads. Each well buffer contained





Peri-Urban areas. Peri-Urban areas were expected to have a greater nitrogen loading rate than Urban due to the use of septic systems (Viers et al., 2012). Loading from these areas can be highly variable depending on septic system density. The estimated loading rate for Peri-Urban could have been affected by the probability that a well had intercepted septic system waste (Bremer and Harter, 2012).

5 Crop groups with a somewhat lower than expected estimated nitrogen loading rate included Manured Forage, Tree Fruit, Cotton, and Nuts: the GNLM study for these crops predicted generally higher loading rates, driven mostly by the difference between applied synthetic fertilizer or manure and harvested nitrogen (see below for more discussion on our model results versus mass balance estimates) (Figure 7).

 The effects seen for estimated loading rates in scenario 1 (CAFO landuse as an explicit group) versus scenario 2 (CAFO
10 lumped with Manured Forage) were a result of loading rates which were estimated based on a linear combination of the proportion of landuse surrounding wells and the measured nitrate concentrations. The proportion of scenario 1 CAFO landuse within well buffers was small (almost all wells had less than 10% CAFO landuse within the buffer, Figure 4), while Manured Forage crops occupied a larger proportion of the area (up to 50% of well buffers were Manured Forage crops Figure 4). Larger proportions of well buffer areas as CAFO landuse or of Manured Forage landuse were related to an increase in nitrate
15 concentration within wells, for CAFO landuse percents above about 0.05 and for Manured Forage landuse above 0.35 (Figure 4). The scenario 1 CIs for CAFO and Manured Forage did not contain overlapping values, and thus we concluded the predicted loading rates for these two groups were statistically significantly different. These results are supported by previous research that estimated loading from manured forage crops to be nearly half of that from lagoons or corrals (van der Schans et al., 2009). While van der Schans et al. (2009) estimated loading from dairy corrals and lagoons to be nearly equal, the GNLM study refers
20 to loading rates for dairy lagoons that are over five times greater than the rate referenced for corral areas (Section 4). It is likely that CAFO landuse near wells in this study have had an acute effect on the amount of nitrate pumped from the wells and that most of the nitrogen loading is due to dairy lagoons (which are lumped with corral area within the CAFO landuse group). The process of merging CAFO landuse with the surrounding Manured Forage landuse reduces the estimated loading rates that would otherwise be specific to CAFO landuse.

25 Previous direct nitrogen loading measurements are highly variable within crop groups, especially for Citrus & Subtropical, Vegetables & Berries, and Cotton (Figure 6). This is the result of several factors including within field crop rotations, variable irrigation and farming nutrient management practices within fields and among farms, and variable measurement methods (Viers et al., 2012). The historical direct nitrogen loading measurements should therefore be interpreted with caution (Viers et al., 2012). Though, many direct measurements are within our model estimated CIs for the specific crop groups with available
30 measurements (Figure 6). The estimated distributions of nitrogen loading therefore represent potential variability of loading rates within a landuse group, as much as they represent uncertainty about the loading rate.

 The discrepancy between some mass balance estimates and the Bayesian model estimates may be due to dilution. Dilution of nitrogen in recharge water is most likely to occur through mixing along the well screen with older, low nitrogen containing, water. Mixing with old water (that recharged prior to the advent of nitrogen fertilizers in the 1930s and 1940s) within well
35 screens could potentially have affected the model estimated loading rates for all crop and landuse groups. Due to the length





of well screens, all domestic well samples contain water of mixed age. A study located near Fresno, CA (within our study area) found that groundwater samples from individual wells contained groundwater with a range of ages, where the range is typically greater than 50 years (Weissmann et al., 2002). Weissmann et al. (2002) attributed the high variance in groundwater residence time within a single well to the heterogeneity within the alluvial aquifer system which produced spatially varying

flow velocities. Weissmann et al. (2002) also reported significant positive skewness (tailing) in the distribution of groundwater ages within individual wells, meaning wells contained some groundwater which was much older than the median age. The authors reported the tailing behavior was due to low hydraulic conductivity units within the aquifer in which slow advection and diffusion dominate the transport process. These results are similar to an earlier study in the Salinas Valley, California (an alluvial aquifer system that, at comparable depth, is similar to the Central Valley and dominated by agriculture) where the

authors found significant dispersion of groundwater ages within simulated groundwater samples (Fogg et al., 1999). Simulated water samples from Fogg et al. (1999) had groundwater ages ranging from 10 years to greater than 500 years. The authors point out that the water pumped from wells in the Salinas Valley was only partially from water that was young enough to be contaminated by nitrate and that this proportion would only increase in the future. Geostatistical analysis of groundwater age tracers from wells sampled in the CV has estimated the depth to the top of well screens pumping pre-modern (age of 60 years

of more) groundwater to be between 30 - 120 m (Visser et al., 2016). According to those results and considering the median depth to bottom of well screen for wells in our study (64 m), a portion of our study wells screened intervals likely penetrate the interface between young and old water. Therefore, mixing within wells with water recharged prior to the intensive use of fertilizers and/or long residence times (tailing effect) could have led to the lower estimates of nitrogen loading for some crops in this study. However, at individual wells, or even across our set of wells, it is difficult to assess the dilution effect with older

water without more detailed analysis of groundwater age throughout the CV and more information on study well screened intervals.

Infiltrating river water in some areas may also dilute otherwise high nitrogen concentrations in recharge water. A study focused on the TLB (within the CV) geospatially related areas near rivers with lower nitrate concentration in wells. The study highlighted areas where major rivers flow into the TLB from the Sierra Nevada Mountains and found that these areas were also

characterized by wells with lower relative nitrate concentrations (Boyle et al., 2012). Wells near rivers may pump a greater amount of low nitrate water from infiltrating river recharge. Boyle et al. (2012) also point out that agricultural areas near rivers are also more likely to receive surface water irrigations. Nitrogen loading from fields receiving low nitrate surface water irrigations is likely to be lower than from fields irrigated with nitrate contaminated groundwater (Boyle et al., 2012). Irrigation water source could have impacted our model estimates and resulted in the lower loading rates compared to mass balance

estimates for some crops.

Denitrification could also play an important role in some differences between our model estimates and mass balance estimates, though we do not suspect widespread regional denitrification. A study focused in the San Joaquin Valley correlated anoxic groundwater conditions to lower nitrate concentrations, but the authors did not attribute this to denitrification (Landon et al., 2011). Instead, Landon et al. (2011) attributed the lower nitrate concentrations in wells with water classified as anoxic to

older groundwater with longer residence times (recharged prior to the intensive use of fertilizers). Landon et al. (2011) did not

(c) Author(s) 2017. CC-BY 3.0 License.





find significant decreases in nitrate concentration in wells due to denitrification. Results of a multi-model averaging approach to estimate oxygen and nitrogen reduction rates in the San Joaquin Valley did estimate denitrification rates to be significant (Green et al., 2016). However, Green et al. (2016) also estimated oxygen reduction rates to be low: median of 0.12 mg L$^{-1}$ yr$^{-1}$. Much of the shallow groundwater in the CV is well-oxygenated: the dissolved oxygen content of Lockhart et al. (2013) study

wells with a measurement (Table 1) was above 5 mg/L on average (Ransom et al., 2016). In addition, Green et al. (2016) found that the estimated rates of oxygen and nitrogen reduction would not protect wells from nitrate contamination, given current nitrogen application rates. We therefore do not expect that denitrification had a significant, overall affect on our model results, but rather may have had an isolated effect on some wells or crops such as Rice. For example, a study on four rice fields in the Sacramento Valley (northern CV) found little to no nitrate leaching below the rice root zone (pore water nitrate levels were

typically below approximately 2.5 mg/L NO$_3$-N). This was attributed to denitrification during the rice growing season when fields are flooded, ammonia volatilization, plant uptake, and crop management practices that contribute to the development of a hardpan layer directly below the rice root zone. This study found very low nitrate concentrations in groundwater wells near rice fields (median value less than 1 mg/L NO$_3$-N) (Liang et al., 2014) and these findings are consistent with our estimates of Rice nitrogen loading. The GNLM mass balance estimates are outside the range of our model predicted CI for Rice: however,

the mass balance estimate may be greater than our model estimate because it does not take into consideration the unmeasured amount of denitrification potentially taking place in saturated clay soils of rice fields.

Our model estimates a greater median groundwater recharge rate ($r$) compared to the prior information from the CVHM model. This is likely because the study wells are concentrated in agricultural areas with greater recharge rates due to irrigation. The CVHM (Faunt, 2009) estimated recharge rates are calculated for the entire Central Valley, including natural areas with few

wells and little agriculture. Many of our study wells were spatially clustered in the Tulare/Kern and Kings subbasins, which had median CVHM estimated recharge rates of 0.21 and 0.35 (m year$^{-1}$), respectively for the 1990 decade. These median rates are near our model estimated median recharge rate of 0.281 (m year$^{-1}$) (Figure 3). Our model estimated GWPA protection factor, $k$ reflects a lower intrinsic susceptibility to aquifer areas outside of GWPA protection zones, likely due to the deeper groundwater there (Figure 2).

## 6    Conclusions

The novel Bayesian tool developed here provides a statistical methodology to relate nitrate measurements in wells to the various types of surrounding landuses as a means to obtain a statistical distribution of nitrate loading rates. The information can provide a better assessment of landuse impacts to water quality based on extensive nitrate data measured in private wells. This method is especially useful absent specific information of individual farm agricultural management practices, specific groundwater

quality, or local hydrogeology in the vicinity of a well. The tool can be used to define high nitrogen loading (high risk) zones (Figure 8). As is apparent from Figures 1 and 8, much of the CV already suffers from or is at risk for serious groundwater contamination by nitrate. Our results indicate much of this contamination has likely originated from Confined Animal Feeding Operations (dairies) and their associated feed crops (with the exception of alfalfa), as well as from Citrus & Subtropical crops





and Vegetable & Berry crops. Yet, spatial correlations between the depth to older water, well construction, direct contamination pathways, groundwater depth, presence of river water recharge and landuse have likely affected the amount of nitrate pumped by wells. Estimates of nitrate loading generally correspond to previous direct measurements or mass balance estimates. On the other hand, Nuts, Cotton, Tree Fruit, and Rice had lower estimated nitrogen loading rates compared to mass balance

estimates. Nuts, Cotton, and Tree Fruit estimates may have been affected by dilution of crop leachate water past the root zone by infiltrating low-nitrate river water or by mixing with older low-nitrate water within the well screen. Land managers should default to the mass balance estimates for those crops. Rice estimates were likely lower than mass balance estimates due to denitrification directly below saturated rice fields, which mass balance estimates did not consider. Estimates of nitrate leaching concentration for particular crop and landuse types, obtained with this tool may not be generalized and transferred to regions

with substantially different climate or geologic, geomorphic, or soils conditions. However, the statistical modeling approach provided here may be applied to other semi-arid, irrigated regions underlain by alluvial aquifers. Our results could potentially be improved with more information on groundwater age and portion of older water pumped by the wells in our study.

## 7 Code availability

Model code may be available upon request.

## 8 Data availability

This dataset is available upon request. Please note well location information, well data group, or well names are not publically available due to confidentiality agreements with well owners.

*Author contributions.* Dr. Thomas Harter designed the research question and experimental structure. Katherine M. Ransom, with assistance from Quinn Barber, prepared the initial well data. Andrew M. Bell processed landuse layers for the dataset. Katherine M. Ransom with

assistance from Mark N. Grote and Arash Massoudieh wrote and ran model code and assessed model results. George Kourakos processed Groundwater Nitrogen Loading Model (GNLM) results for use in this study. Katherine M. Ransom prepared the manuscript text and figures.

*Competing interests.* The authors declare that they have no conflict of interest

*Acknowledgements.* We are extremely grateful to Mark N. Grote and Arash Massoudieh for their help with model development and assessment, our work would not have been possible without you. Thank you to Jo Ann M. Gronberg for preparing figure map templates of the

Central Valley, California and to Claudia C. Faunt and Jon Traum for providing and processing the CVHM data for our initial estimates of groundwater recharge rates. Partial funding for this work was provided through the State Water Resources Control Board (SWRCB) Agreement No. 09-122-250, and through a grant from the California Department of Agriculture Fertilizer Research and Education Program, project numbers 11-0301 and 15-0454.





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





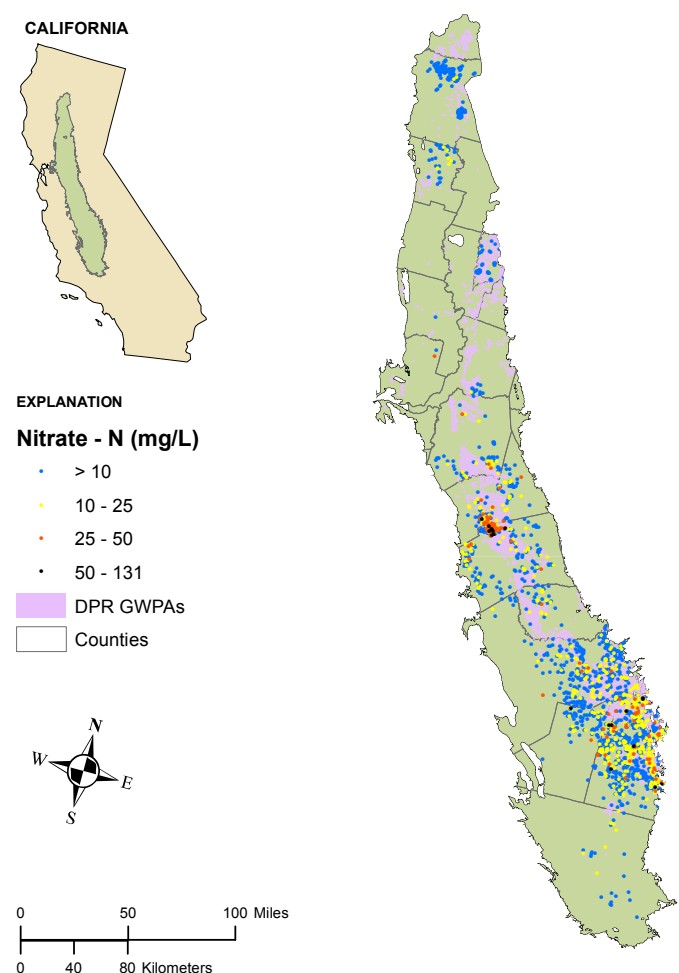

**Figure 1.** Study well locations color coded by nitrate (NO$_3$-N mg/L) value overlain with CDPR GWPA zones (runoff and leaching undifferentiated).





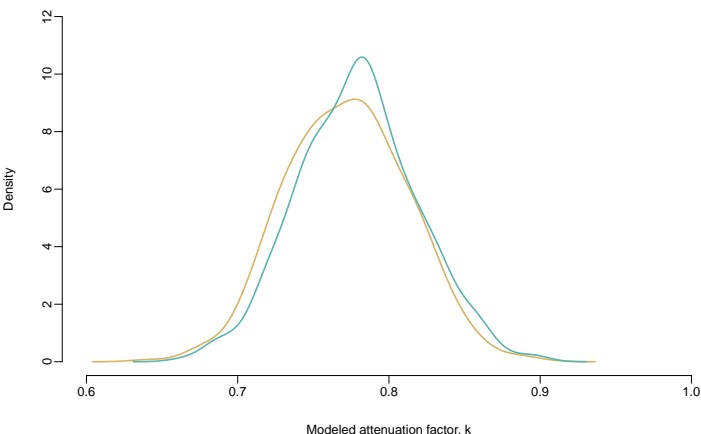

**Figure 2.** Posterior probability density for the non-GWPA attenuation factor, $k$, for scenario 1 (tan) and 2 (teal).

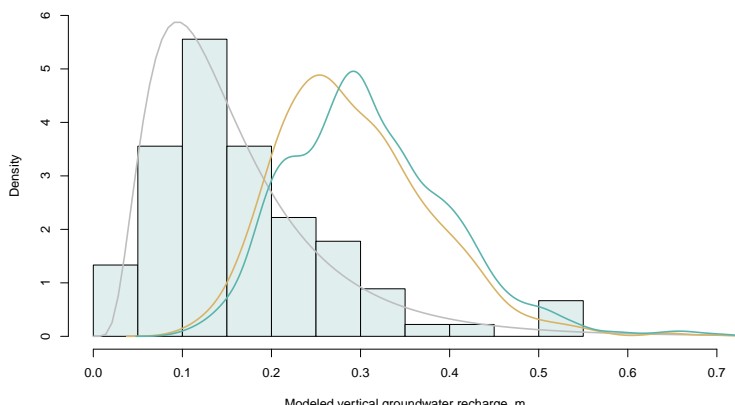

**Figure 3.** CVHM estimated annual vertical groundwater recharge (grey bars), log-normal prior probability density input to nitrate loading model (grey line) and posterior probability density for the nitrate loading model estimated annual recharge rate for scenario 1 (tan) and 2 (teal).





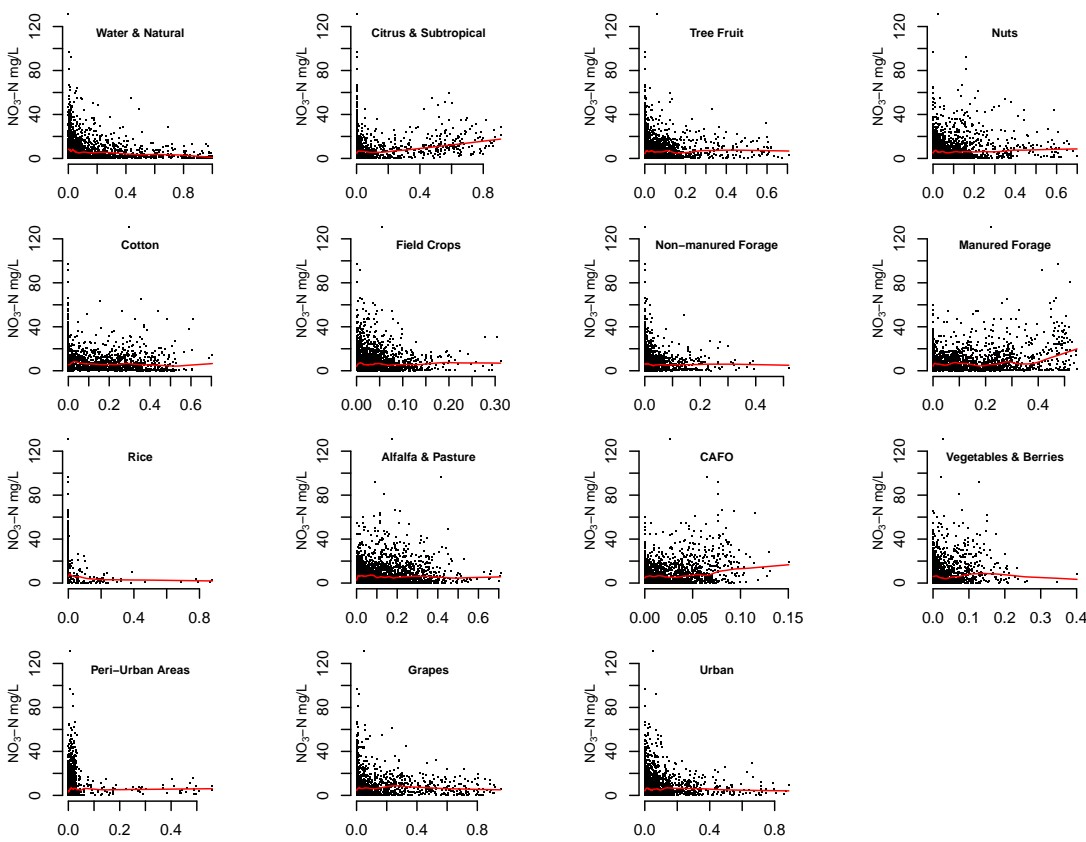

**Figure 4.** Scatterplot of proportion of landuse within each well buffer versus well nitrate concentration for each of the 15 landuse or crop groups in scenario 1. The red line is a locally weighted scatterplot smoothing line (Cleveland, 1979). Note that each plot shows nitrate concentrations in all 2149 wells and the x-axis is scaled differently among subplots for better resolution.





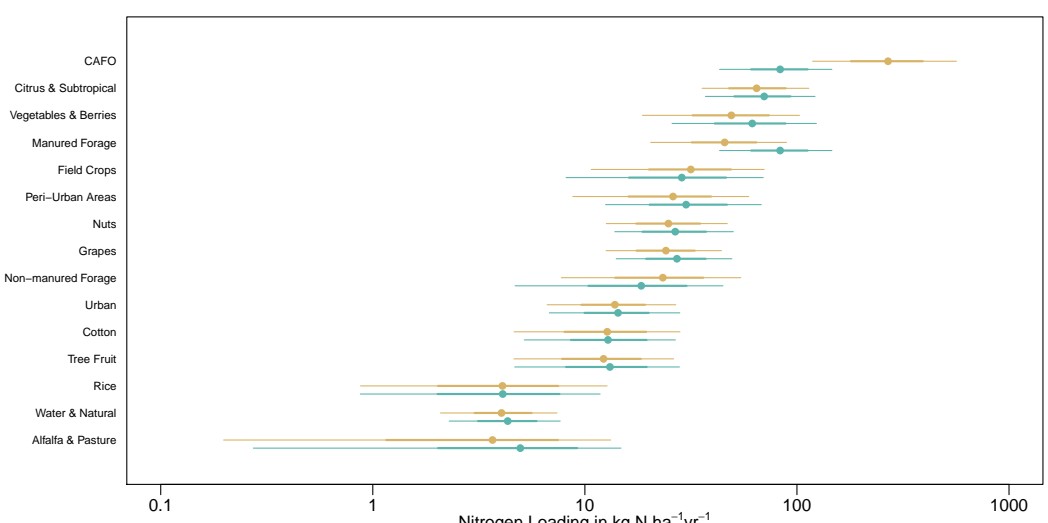

**Figure 5.** Estimated nitrogen loading for scenario 1 (tan) and 2 (teal). Results from scenario 2 for the manured forage & CAFO group were plotted twice: once under scenario 1 CAFO and again under scenario 1 manured forage. Thinner lines are 95% credibility intervals, thicker lines are 68% credibility intervals, and the dot is the median estimated value.





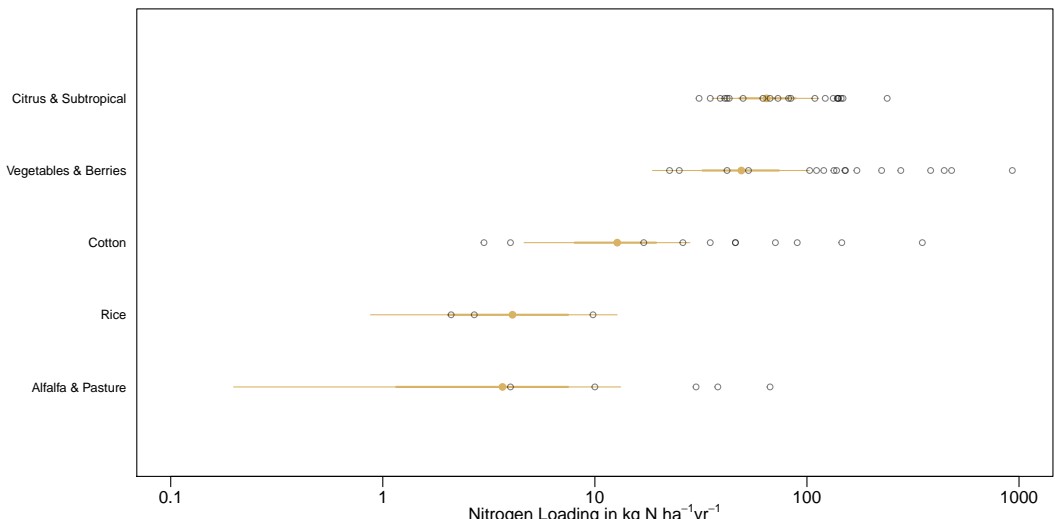

**Figure 6.** Scenario 1 results for selected crop groups (tan) with historical direct measurements of nitrogen loading from California (grey circles, (Devitt et al., 1976; Embleton et al., 1979; Letey et al., 1977; Pratt et al., 1972; Pratt and Adriano, 1973; Adriano et al., 1972; Allaire-Leung et al., 2001; Liang et al., 2014). Thinner lines are 95% credibility intervals, thicker lines are 68% credibility intervals, and the dot is the median estimated value.

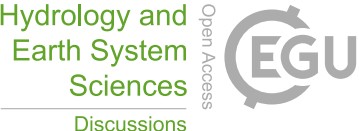



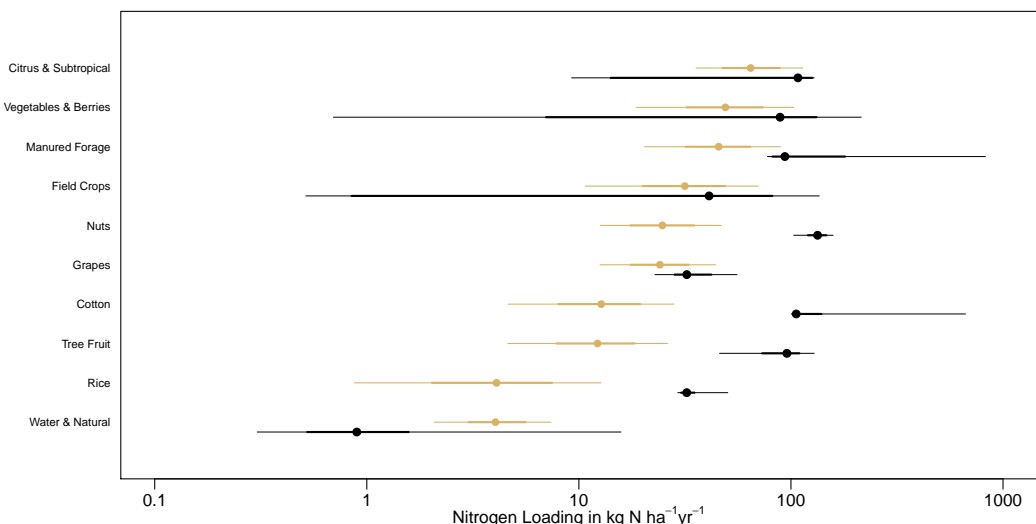

**Figure 7.** Scenario 1 results for selected crop groups (tan) with results from the Groundwater Nitrogen Loading Model (GNLM) (Viers et al., 2012; Rosenstock et al., 2013) (black). Thinner lines are 95% credibility intervals, thicker lines are 68% credibility intervals, and the dot is the median estimated value.





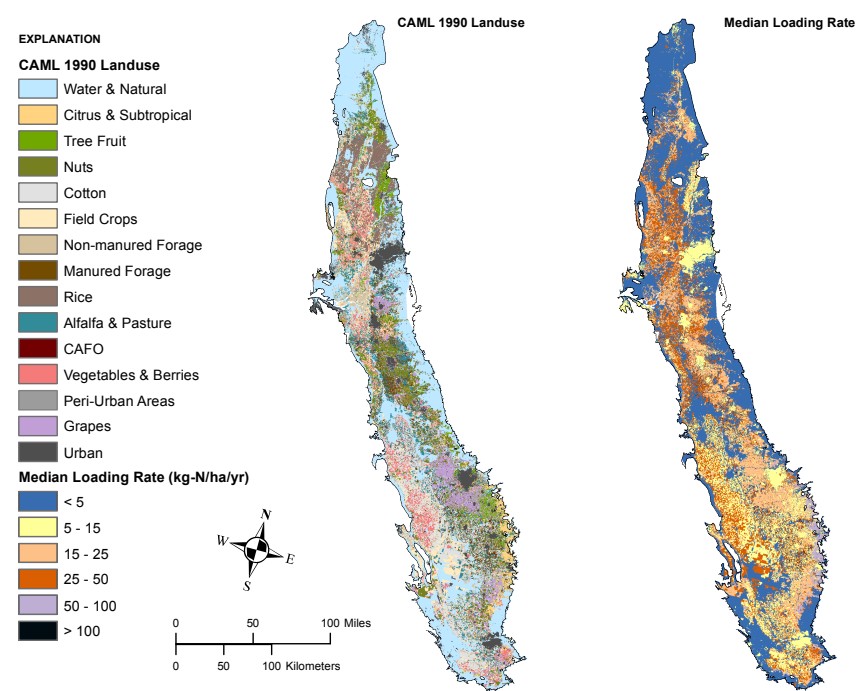

**Figure 8.** CAML landuse for the 15 landuse groups in scenario 1 (left side) and the same landuse groups keyed to the median estimated nitrogen loading rate in kg N ha$^{-1}$ yr$^{-1}$ for the corresponding group from Table 2 (right side).



**Table 1.** Original data source, number of wells, and well location method for private wells included in final database (2149 wells total).

| Dataset Group | Dataset Subgroup | Number of Wells | Location Method/Accuracy |
|---|---|---|---|
| CASTINGS, CVRWQCB Fresno Office | Private wells on dairies | 361 | Located at reported coordinates of the dairy, the reported street address of the dairy, or the centroid of dairy parcel(s) (single, multiple adjacent parcels, or centroid of multiple non-adjacent parcels) (Boyle et al., 2012). |
| CASTINGS | GAMA Domestic Tulare County | 134 | Well locations randomly offset by 1/2 mile from true location (Boyle et al., 2012). |
| CASTINGS | Department of Pesticide Regulations (DPR) | 62 | Located at the centroid of the United States Public Land Survey System (PLSS) section (approximately 1 $mi^2$) in which the well resides (within 1/2 mile of the actual well location) (Boyle et al., 2012). |
| CASTINGS | Fresno County | 295 | Located at street address reported on well logs or centroid of the reported Assessor's Parcel Number (APN) (Boyle et al., 2012). |
| CASTINGS | The U.S. Geological Survey's (USGS) National Water Information System (NWIS) | 17 | Unknown (Boyle et al., 2012). |
| CASTINGS | Tulare County Environmental Health | 437 | Located at centroid of the reported APN (Boyle et al., 2012). |
| Lockhart et al. (2013) | None | 200 | Geographic coordinates digitized with imagery from Google Earth (Lockhart et al., 2013). |
| GAMA Domestic | Tehama, El Dorado, and Yuba Counties | 253 | Well locations randomly offset by 1/2 mile from true location. |
| CVRWQCB Rancho Cordova Office | Private wells on dairies | 390 | Geocoded using street address. |





**Table 2.** Scenario 1 95% credibility interval bounds and median modeled nitrogen loading rates by group.

| Group | Median | Lower Bound | Upper Bound |
|---|---|---|---|
| Water & Natural | 4.03 | 2.11 | 7.43 |
| Citrus & Subtropical | 64.60 | 36.87 | 113.92 |
| Tree Fruit | 12.26 | 4.85 | 26.68 |
| Nuts | 25.12 | 13.28 | 46.91 |
| Cotton | 12.74 | 5.17 | 28.42 |
| Field Crops | 31.65 | 9.84 | 71.65 |
| Non-manured Forage | 23.89 | 8.52 | 54.21 |
| Manured Forage | 44.85 | 20.50 | 92.41 |
| Rice | 4.11 | 0.88 | 12.68 |
| Alfalfa & Pasture | 3.51 | 0.20 | 13.49 |
| CAFO | 272.82 | 121.03 | 569.53 |
| Vegetables & Berries | 48.84 | 19.44 | 102.67 |
| Peri-Urban Areas | 25.88 | 8.62 | 59.62 |
| Grapes | 24.00 | 12.70 | 42.46 |
| Urban | 13.69 | 6.78 | 26.67 |





**Table 3.** Scenario 2 95% credibility interval bounds and median modeled nitrogen loading rates by group.

| Group | Median | Lower Bound | Upper Bound |
|---|---|---|---|
| Water & Natural | 4.32 | 2.23 | 7.54 |
| Citrus & Subtropical | 69.00 | 37.01 | 116.52 |
| Tree Fruit | 12.98 | 4.73 | 26.22 |
| Nuts | 26.66 | 13.83 | 46.82 |
| Cotton | 12.75 | 5.29 | 25.83 |
| Field Crops | 28.83 | 9.42 | 71.36 |
| Non-manured Forage | 18.07 | 4.44 | 44.89 |
| Manured Forage & CAFO | 83.42 | 43.22 | 146.64 |
| Rice | 4.19 | 0.90 | 11.79 |
| Alfalfa & Pasture | 4.96 | 0.28 | 13.51 |
| Vegetables & Berries | 61.35 | 25.73 | 122.48 |
| Peri-Urban Areas | 30.14 | 12.31 | 68.64 |
| Grapes | 27.07 | 14.03 | 49.30 |
| Urban | 14.21 | 6.51 | 27.89 |