# Peer review of "A Bayesian Approach to Infer Nitrogen Loading Rates from Crop and Landuse Types Surrounding Private Wells in the Central Valley, California"

_Hydrology and Earth System Sciences, 2017_

## Referee Comment (RC1) · Anonymous Referee #1 · 20 Mar 2017

This study used nitrate measurements from many wells in the Central Valley, CA, to estimate nitrogen loading rate distributions for different crop and landuse types, using a Bayesian regression model. After reading the manuscript, I think the author should address the following major concerns. The author mentioned that they developed a novel Bayesian regression model, but since the whole manuscript lacks the introduction to previous statistical methods applied for N loading estimation and the reference of the application of Bayesian method related to the topic, it would be difficult for the reader to sense what the novelty is. The method section describes a lot about the site and data, leaving the statistical method vague and missing the implementation of the Bayesian

method and the details (equations and descriptions) of the initial approximation and MCMC method for the posterior distributions. Overall, the manuscript is well-written except for a few results and conclusions not following the rigor of scientific standards (see specific comments). Moreover, some of the results (figures and tables) were not well organized or presented as pointed out in the specific comments below.

Specific comments:

The abstract is too general, and most of the contents seem to belong to the introduction section. Although the author claimed the development of the Bayesian regression model, the abstract did not emphasize the finding of this work using the model. And the focus of the work is clearly not the model development.

P2L8, "Drinking water with nitrate concentrations above background levels of near 1 mg/L ..." What are the 'background levels'? They were not mentioned before.

P2L19∼20, What is the source of these numbers? Any reference?

As the author mentioned they developed a novel Bayesian regression model, the introduction should describe the current research status of statistical methods used for the related topic, and whether the Bayesian method has been applied in this area. Otherwise, it is hard to tell what the scientific contribution is of this work.

P3L17, 'Spring 2011 depth to groundwater ...' meaning in the Spring of 2011?

P4L23∼24, how was the database filtered? why did you use median value, any reason for that?

P4L26, '4.4268', how to calculate this 'mass ratio'?

P4L29, 'ransom' –> random?

P5L10∼11, what are the soil properties that lead to the same behavior of pesticide and nitrate contamination? Or just simply because they are both hydrophilic?

P5L13, 'raster' –> raster image file?

P5L14, briefly introduce the data sources of CAML.

P5L34, why is '2.4 km'? How to calculate? And why is '0.30' m per year?

P6L13, 'occuring' –> occurring

P6L5~17, has CVHM ever been fully tested for the research area? How accurate is this hydrologic model? Any reference?

P7L13, 'reflect' –> reflects

P7L9~16, what are the 'location' and 'scale' parameters? Student t-distribution has only one parameter, the degree of freedom. Present the equation for the distribution here.

P8L5, '(1)' –> Figure 1?

P8L6, 'non-parametric Kruskal-Wallis test' should be described in the method section.

P8L25~28, 'Pearson goodness-of-fit' and 'standardized Pearson goodness-of-fit' should also be described in the method section.

P9L1~4, this sentence is too long and unclear.

P9L29, 'a' –> as

P9L31, 'are greater than', why is that? If you wrote something, then discuss it accordingly. Or do not mention it.

It should be better to put all comparisons with references in the discussion section, and no reference appears in the results section.

P10L16, what is the meaning to put the parenthesis and the statement about alfalfa here?

P11L13, missing the punctuation in the parenthesis.

P14L1, 'spatial correlation' does not appear in either results or discussion sections, how is it shown in the conclusion?

Figures

Fig. 1 was not referred throughout the entire manuscript.

Figs. 2, 3, 5, 6, and 7 need the legends.

Fig. 4, what are the x-axes?

Figs. 5 and 6, if data were not plotted in log scale, numbers in Tables 2 and 3 are repeated. Readers can receive the same information from the figures alone. Figs 5 and 6 can be combined as one. Figs 5, 6, and 7 are hard to read, suggest to change the style to bar plot, with landuse types on the x-axis and N loading on the y-axis.

---

## Referee Comment (RC2) · Anonymous Referee #2 · 2 Jul 2017

Review comments to hess-2017-39: A Bayesian approach to infer nitrogen loading rates from crop and landuse types surrounding private wells in the Central Valley, California by Katherine M. Ransom et al.

This paper presents a Bayesian regression model that provides a statistical methodology to relate nitrate measurements in wells to the various types of surrounding landuses as a means to obtain a statistical distribution of nitrate loading rates. The study is focused in the Central Valley, California, USA, an intensively farmed region with high agricultural crop diversity.

[Figure]

This method is especially useful absent specific information of individual farm agricultural management practices, specific groundwater quality, or local hydrogeology in the vicinity of a well. The tool can be used to define high nitrogen loading (high risk) zones. Authors have done interesting work. This paper has a good potential to be published in the journal. However, there are some significant issues, listed below, which need to be addressed before it is ready for publication. 1. Abstract section: Please rewrite this section, and focus more in what you have done including the study results in the manuscript.

2. Introduction section: Introduce more on Bayesian statistical models and why authors developed such types of models.

3. Combining the Results section and Discussion section. When each picture is shown, we would like to see the description for the picture and why this phenomenon happens. So it is better to combine the Results and Discussion sections.

4. In page 3, line 17-18: Spring 2011 depth to groundwater ranged from 10 feet below ground surface (bgs) in the northern section of the CV to 670 feet (bgs) in the southern portion of the CV (DWR, 2011).

5. In page 5, line 15-16: Insert "it" between "because" and "is".

6. In page 6, line 29: Delete blank space before the "where".

---

## Author Comment (AC1) · 29 Sep 2017

We thank the reviewer for his or her comments on our work. We have attached a revised PDF incorporating our response to the comments. Please see specific responses below.

Anonymous Referee #1

This study used nitrate measurements from many wells in the Central Valley, CA, to estimate nitrogen loading rate distributions for different crop and landuse types, using

a Bayesian regression model. After reading the manuscript, I think the author should address the following major concerns. The author mentioned that they developed a novel Bayesian regression model, but since the whole manuscript lacks the introduction to previous statistical methods applied for N loading estimation and the reference of the application of Bayesian method related to the topic, it would be difficult for the reader to sense what the novelty is. The method section describes a lot about the site and data, leaving the statistical method vague and missing the implementation of the Bayesian method and the details (equations and descriptions) of the initial approximation and MCMC method for the posterior distributions. Overall, the manuscript is well-written except for a few results and conclusions not following the rigor of scientific standards (see specific comments). Moreover, some of the results (figures and tables) were not well organized or presented as pointed out in the specific comments below.

Specific comments:

The abstract is too general, and most of the contents seem to belong to the introduction section. Although the author claimed the development of the Bayesian regression model, the abstract did not emphasize the finding of this work using the model. And the focus of the work is clearly not the model development. Response: We agree that the abstract is too general and did not focus on the findings of the work. We have rewritten the abstract to focus more on the need for the study, the novel aspects, and include several sentences to mention specific findings.

P2L8, "Drinking water with nitrate concentrations above background levels of near 1 mg/L ..." What are the 'background levels'? They were not mentioned before. Response: We have inserted a reference for the background level of about 2 mg/L.

P2L19_20, What is the source of these numbers? Any reference? Response: We have inserted the appropriate references here.

As the author mentioned they developed a novel Bayesian regression model, the introduction should describe the current research status of statistical methods used for

the related topic, and whether the Bayesian method has been applied in this area. Otherwise, it is hard to tell what the scientific contribution is of this work. Response: Statistical methods have not been used, to our knowledge, to estimate nitrogen loading rates to groundwater. In the introduction we discuss how the previous work in this field is highly limited and mostly based on field studies. We have included some additional description in the Introduction section on the use of Bayesian statistical methods for estimating loading coefficients to surface water, and for nitrate source apportionment. In addition, we include text to point out that Bayesian methods have not been used previously to estimate nitrogen loading rates to groundwater. We have also included an additional section (new section 3.1) titles "Conceptual model" which explains in more detail nitrate transport modeling and the rational behind the Bayesian method.

P3L17, 'Spring 2011 depth to groundwater ...' meaning in the Spring of 2011? Response: That is correct and we feel this description is adequate, we have left it as is.

P4L23_24, how was the database filtered? why did you use median value, any reason for that? Response: We subset the database in the R statistical program to select only the more recent records (from between 2000-2015). Median value was used to prevent giving more weight to wells that had been sampled more than once by representing them, and the associated landuse, with many samples (several wells had yearly or monthly sampling). We believe these methods to be well described in the paper and have left the description as is.

P4L26, '4.4268', how to calculate this 'mass ratio'? Response: It is the molar mass of nitrate divided by the molecular weight of nitrogen. This is a standard conversion and we therefore removed the reference to the conversion factor and just state that the values were converted.

P4L29, 'ransom' –> random? Response: We have corrected this typo.

P5L10_11, what are the soil properties that lead to the same behavior of pesticide and

nitrate contamination? Or just simply because they are both hydrophilic? Response: We added clarifying text. Shallow water table, short residence time in the vadose zone, and low reactivity in the aquifer materials are key risk factors captured by the GWPA designation used here. Implicitly, but irrelevant to the choice here, it is indeed hydrophilic pesticides that would be the most likely to contaminate a well (as opposed to hydrophobic pesticides). P5L13, 'raster' –> raster image file? Response: Yes, and we have changed the word "raster" here to "raster image file".

P5L14, briefly introduce the data sources of CAML. Response: Text added.

P5L34, why is '2.4 km'? How to calculate? And why is '0.30' m per year? Response: These are representative values for the Central Valley. Text was rearranged to further clarify and references are included.

P6L13, 'occuring' –> occurring Response: We have corrected this typo.

P6L5_17, has CVHM ever been fully tested for the research area? How accurate is this hydrologic model? Any reference? Response: CVHM is a well established groundwater model of the Central Valley. The reference (Faunt, 2009) includes calibration data and established the overall level of accuracy of the model.

P7L13, 'reflect' –> reflects Response: We have corrected this typo.

P7L9_16, what are the 'location' and 'scale' parameters? Student t-distribution has only one parameter, the degree of freedom. Present the equation for the distribution here. Response: We have used the form of the t-distribution containing the location and scale that is more common to modeling approaches. The standardized form of the t-distribution to which you refer, containing only the degrees of freedom parameter, can be converted to the form with the location and scale by the following equation: $f(x;a,b) = (1/b)f((x-a)/b;0,1)$ Where $f()$ is the standardized t-distribution, a is the location and b is the scale. This gives the equation found here: https://www.mathworks.com/help/stats/t-location-scale-distribution.html?requestedDomain=www.mathworks.com. We have included a reference to the JAGS user guide in the text, to refer the reader to the form of this probability distribution used in the study.

P8L5, '(1)' –> Figure 1? Response: Yes, and we have corrected this typo.

P8L6, 'non-parametric Kruskal-Wallis test' should be described in the method section. Response: We have added the following two sentences to the Methods section under Well and Landuse data: The non-parametric Kruskal-Wallis statistical test was performed on the nitrate values for wells in each of the two groups (GWPS versus non-GWPA wells). The Kruskal-Wallis test is a ranked one-way analysis of variance which tests whether two groups of values should be considered independent or from the same distribution.

P8L25_28, 'Pearson goodness-of-fit' and 'standardized Pearson goodness-of-fit' should also be described in the method section. Response: This is described at the very end of the Methods section under subsection Statistical methods.

P9L1_4, this sentence is too long and unclear. Response: We have made this two sentences and edited for clarity.

P9L29, 'a' –> as Response: This typo has been corrected.

P9L31, 'are greater than', why is that? If you wrote something, then discuss it accordingly. Or do not mention it. Response: In an effort to keep the results and discussion sections separate, this is discussed in discussion on page 13.

It should be better to put all comparisons with references in the discussion section, and no reference appears in the results section. Response: We agree and have moved the results paragraphs 6, 7, and 8 to the discussion section and edited for clarity.

P10L16, what is the meaning to put the parenthesis and the statement about alfalfa here? Response: We include this statement to describe why alfalfa does not need nitrogen applications and is expected to therefore have low nitrogen loading rates.

P11L13, missing the punctuation in the parenthesis. Response: This typo has been corrected. P14L1, 'spatial correlation' does not appear in either results or discussion sections, how is it shown in the conclusion? Response: We agree here and have changed the phrase to "interactions".

Figures Fig. 1 was not referred throughout the entire manuscript. Response: This has been corrected above.

Figs. 2, 3, 5, 6, and 7 need the legends. Response: We have added legends to these figures.

Fig. 4, what are the x-axes? Response: We have edited this figure to include x-axis labels for each plot. Each plot has the same value for the x-axis: the proportion of well buffer area as the plotted land use.

Figs. 5 and 6, if data were not plotted in log scale, numbers in Tables 2 and 3 are repeated. Readers can receive the same information from the figures alone. Figs 5 and 6 can be combined as one. Figs 5, 6, and 7 are hard to read, suggest to change the style to bar plot, with landuse types on the x-axis and N loading on the y-axis. Response: We have reformatted Figure 5 to display the estimated probability densities, with each landuse group a separate plot. This has removed the need for a log scale. We see the data presented in Tables 2 and 3 as useful to the reader and therefore, we have combined Tables 2 and 3. We are unable to combine Figures 5 and 6 due to scaling issues (some direct measurements of nitrogen loading are much greater than our model estimates, and are therefore difficult to display nicely on one plot). We have therefore left Figure 5 and 6 separate, but for Figure 6 we have spread out the data a bit, included more tic marks on the x-axis, and included a legend. For Figure 7, we have also spaced the data out for ease of readability, added tic marks and labels, and included a legend.

Please also note the supplement to this comment:

https://www.hydrol-earth-syst-sci-discuss.net/hess-2017-39/hess-2017-39-AC1-supplement.pdf

---

## Author Comment (AC2) · 29 Sep 2017

We thank the reviewer for his or her comments on our work. Please see the attached revised PDF for incorporation of the comments. See specific responses to comments below.

Review comments to hess-2017-39: A Bayesian approach to infer nitrogen loading rates from crop and landuse types surrounding private wells in the Central Valley, California by Katherine M. Ransom et al. This paper presents a Bayesian regression model

[Figure]

that provides a statistical methodology to relate nitrate measurements in wells to the various types of surrounding landuses as a means to obtain a statistical distribution of nitrate loading rates. The study is focused in the Central Valley, California, USA, an intensively farmed region with high agricultural crop diversity. This method is especially useful absent specific information of individual farm agricultural management practices, specific groundwater quality, or local hydrogeology in the vicinity of a well. The tool can be used to define high nitrogen loading (high risk) zones. Authors have done interesting work. This paper has a good potential to be published in the journal. However, there are some significant issues, listed below, which need to be addressed before it is ready for publication.

1. Abstract section: Please rewrite this section, and focus more in what you have done including the study results in the manuscript. Response: We agree with both reviewers that the abstract did not focus enough on the results and findings of the study. We have rewritten the abstract to focus much more on the specific findings and results.

2. Introduction section: Introduce more on Bayesian statistical models and why authors developed such types of models. Response: We have included additional information on previous Bayesian studies in the introduction section. We are not currently aware of any study that has used Bayesian methods to estimate nitrate loading rates to groundwater. We have also included an additional section (new section 3.1) titles "Conceptual model" which explains in more detail nitrate transport modeling and the rational behind the Bayesian method.

3. Combining the Results section and Discussion section. When each picture is shown, we would like to see the description for the picture and why this phenomenon happens. So it is better to combine the Results and Discussion sections. Response: Due to the complicated nature of the discussion of our model results, we feel it is really best to leave them separate. We compare various parts of the model results with several previous studies and feel combining the results and discussions would lead to disorganization of the presentation as not all results are compared to all studies. This
way we can present all the results up front, and follow up with more detailed specific discussions of model results separately.

4. In page 3, line 17-18: Spring 2011 depth to groundwater ranged from 10 feet below ground surface (bgs) in the northern section of the CV to 670 feet (bgs) in the southern portion of the CV (DWR, 2011). Response: We have left the parentheses off bgs here, as it represents an abbreviation and is used later on in the manuscript.

5. In page 5, line 15-16: Insert "it" between "because" and "is". Response: This typo has been corrected.

6. In page 6, line 29: Delete blank space before the "where". Response: This has been corrected.

Please also note the supplement to this comment:
https://www.hydrol-earth-syst-sci-discuss.net/hess-2017-39/hess-2017-39-AC2-supplement.pdf

**Supplement:**

[revised manuscript text omitted]

on shallow private wells for drinking and household purposes. Private wells are not regulated in California and it is difficult to know how many may be contaminated by nitrate: a significant portion of the CV has been estimated to contain shallow groundwater with elevated nitrate concentration (Nolan et al., 2014; Lockhart et al., 2013; Ransom et al., 2017).

**3 Methods**

**3.1 Conceptual model**

To use well nitrate measurements for estimating nitrate (as nitrogen) losses to groundwater from specific landuses, we employ a stochastic Bayesian inference model that we derive from physically-based concepts about groundwater dynamics near a pumping well, the location of its source area, and the overlying crop or landuse type. Each crop and landuse type is characterized by water mass flux (recharge rate) and an associated nitrate mass flux to groundwater.

Domestic wells typically supply a single household. The average per household water consumption in California is $1.2 \times 10^3$ m$^3$ yr$^{-1}$ (1 acft yr$^{-1}$, 2 L min$^{-1}$, 0.5 gpm). At a recharge rate of 0.3 m yr$^{-1}$ (1 ft yr$^{-1}$), for example, the source area is approximately 0.4 ha (1 acre). In productive aquifer systems such as that of the CV, this source area typically has a long but very narrow shape (Horn and Harter, 2009). At any time, the water produced by the well is a mixture of water retrieved from a continuous range of depth along the screen of the well representing contributions from across the source area:

$$Q_i(t) = \int\limits_x \int\limits_y q[x, y, t - \tau_i(x, y, t)] dx dy \tag{1}$$

where $Q_i(t)$ is the pumping rate at well $i$ at time $t$, $q()$ is the recharge rate at location $x, y$ at time $(t - \tau)$ (vertical length per time), $\tau_i(x, y, t)$ is the age of the water contributing from point $x, y$ at time $t$. A well's nitrate concentration is therefore also a mixture of nitrate contributions across the source area:

$$C_i(t) = \int\limits_x \int\limits_y \frac{m[x, y, t - \tau_i(x, y, t)]}{q[x, y, t - \tau_i(x, y, t)]} dx dy \tag{2}$$

where $C_i(t)$ is the measured nitrate concentration in well $i$ at time $t$, and $m()$ is the nitrate mass flux associated with the recharge $q()$. The nitrate mass flux is defined by:

$$m[x, y, t - \tau_i(x, y, t)] = c[x, y, t - \tau_i(x, y, t)] * q[x, y, t - \tau_i(x, y, t)] \tag{3}$$

where c is the nitrate concentration in recharge. Note that the above equation implicitly accounts for variability of water and nitrate flux across the surface of the screen.

Under ideal conditions, if $m(x, y, t - \tau)$ is a constant but unknown quantity for a crop or other mappable landuse type, if the source area is known, and if the recharge rate $q(x, y, t - \tau)$ is known, then measurements of $Q_i(t)$ and $C_i(t)$ at N wells is

sufficient to compute $m(x,y,t-\tau)$ and, hence, $c(x,y,t-\tau)$. Writing Equation 2 for each of N wells, where N is the number of crops and landuse types, yields N equations with N unknowns that could be solved exactly.

In practice, neither of these quantities is well known. Despite the availability of regional groundwater flow maps (DWR, 2011) and models (Faunt, 2009), uncertainty about the location of the source area arises from hydrological heterogeneity and largely unknown spatio-temporal variability in large scale groundwater pumping near sampled wells. These factors can greatly alter groundwater flow and thus the source area of a well, often seasonally. Previous studies have therefore used a circular "buffer" zone around each well as an approximation of the well source area (Burow et al., 1998a; McLay et al., 2001; Kolpin, 1996; Lockhart et al., 2013). Circular well buffers have been shown to be reasonable approximations of the potential well source area when the actual contributing source area is unknown (Johnson and Belitz, 2009).

The exact contributions $q(x,y,t-\tau)$ and $m(x,y,t-\tau)$ are also not known. While the crop or landuse type at location $x,y$ is here thought to have a major influence on the magnitude of $m(x,y,t-\tau)$, the specific loading at any location associated with a specific crop or landuse type exhibits within-crop spatio-temporal variability, which arise from variation in farming practices (farm to farm variation) and from hydrogeologic, pedologic, and agricultural practice variability (within farm variation).

When linking well nitrate to nitrate (as nitrogen) mass flux from specific crops and other landuse types, there are therefore three sources of uncertainty: First, the actual source area is but a small sliver of the buffer region. Hence, the contributing sources are uncertain, but constrained by the crop and other landuse type composition within the buffer. Second, the total contribution to mass loading from each crop within the source area may vary due to various factors related to the spatio-temporal variability of agricultural and environmental conditions within the crop area contributing to $Q_i(t)$. Third, the recharge rate $q(x,y,t-\tau)$ is unknown.

The following sections explain the sources of well nitrate data, the computation of the buffer radius, and the mapping of crop and other landuse types that we employed here to demonstrate the usefulness of this approach (Section 3.2). We then explain the Bayesian inference methodology that we employed to account for the aforementioned uncertainties (Section 3.3).

**3.2 Well and landuse data**

A database of nitrate measurements from private wells located in California was compiled from several data sources. The California Ambient Spatio-Temporal Information on Nitrate in Groundwater (CASTING) includes nitrate measurements from private supply, public supply, irrigation, and monitoring wells (Boyle et al., 2012). We selected all well measurements from the CASTING database from supply wells (not monitoring wells) designated as private. We selected samples from private wells because they are typically more shallow than public supply or irrigation wells and are not purposefully located near sources of contamination as are monitoring wells. Private well samples within the CASTING database originated from several sources including the Central Valley Regional Water Quality Control Board (CVRWQCB) Fresno Office dairy domestic wells monitoring data (sampled for nitrate as a part of the Dairy General Order regulations for dairy facilities in the CV), the California Department of Pesticide Regulation (CDPR), Fresno County, the United States Geological Survey (USGS), Tulare County Environmental Health (TCEH), and the State Water Resources Control Board (SWRCB) Groundwater Ambient Monitoring Assessment (GAMA) Domestic Wells Project in Tulare County. We expanded the original database, which was geographically

limited to the southern CV, to include the data from the same data sources for the entire CV. Also, additional private well samples from the following data sources were added to the CASTING database:

- a set of private wells previously sampled for nitrate as a part of the "Proposition 50 Long Term Risk of Groundwater and Drinking Water Degradation from Dairies and Other Nonpoint Sources in the San Joaquin Valley", funded by the State Water Resources Control Board (SWRCB) (Lockhart et al., 2013) (200 wells total, sampled between 2010-2011),

- additional SWRCB GAMA private wells for Tehama, El Dorado, and Yuba county project areas (GAMA Domestic Well Project, http://www.waterboards.ca.gov/gama/domestic_well.shtml) downloaded from the GeoTracker GAMA online database (http://www.waterboards.ca.gov/gama/geotracker_gama.shtml), and

- CVRWQCB Rancho Cordova Office dairy domestic wells monitoring data provided by the CVRWQCB office.

Records in the database collected between the years 2000 to 2015 were selected. Locations with data collected in multiple years were assigned the median nitrate value of all the recorded measurements in order to prevent multiple samples of the same well and associated landuse. Prior to median aggregation, non-detect nitrate values were replaced with the detection limit and zero values were replaced with the most common detection limit of 2.21 mg/L $NO_3$-$NO_3$. All nitrate measurements were then converted to $NO_3$-N. When geographic coordinates (latitude and longitude) of the private wells in the dairy monitoring program were not available, the wells were located using the dairies street address and placed at the centroid of a dairy's land parcels. The methods for locating the wells varied for each of the other data sources including geographic coordinates, geocoded addresses, offsets by a random small distance, United States Public Land Survey System (PLSS) section, and Assessor's Parcel Number (APN) (Table 1). Due to the well location methods, many wells had overlapping locations. Where multiple wells were geolocated to a single location, a single well was chosen at random to represent that location. Wells outside of the alluvial aquifer system boundary were excluded from the analysis. The final nitrate database had a total of 2149 wells.

Intrinsic aquifer properties were evaluated as an indicator for additional risk for or protection from nitrogen contamination. Here we choose a simple binary indicator: California Department of Pesticide Regulations (CDPR) Groundwater Protection Areas (GWPAs) are 1.6 km square sections that are vulnerable to the leaching of pesticides and are defined by the following criteria: previous detections of pesticides in that section, contains coarse soils and a depth to groundwater less than 21 m, or contains runoff-prone soils and depth to groundwater less than 21 m (California Department of Pesticide Regulation, 2017a). These zones are either vulnerable to contamination due to non-point source leaching of irrigation water "leaching GWPAs" or direct flow paths through hardpan soils (ditches, dry wells, poorly sealed wells, "runoff GWPAs") (California Department of Pesticide Regulation, 2017b). The properties that lead to vulnerabilities from pesticide contamination - shallow depth to groundwater, short residence time in the vadose zone, low reactivity of aquifer sediments - also increase the possibility of nitrate contamination. Wells located within a GWPA zone were attributed as being an indicator for increased risk of nitrate contamination. The non-parametric Kruskal-Wallis statistical test was performed on the nitrate values for wells in each of the two groups (GWPS versus non-GWPA wells). The Kruskal-Wallis test is a ranked one-way analysis of variance which tests whether two groups of values should be considered independent or from the same distribution.

Landuse surrounding wells was analyzed using the California Augmented Multi-Source Landuse (CAML) 50 m resolution raster image file for the year 1990 (Hollander, 2013). CAML was developed from various data sources delineating various natural vegetation types, farmland, and urban areas for five periods of five years each centered on 1945, 1960, 1975, 1990, and 2005. The data sources included the California Department of Conservation Farmland Mapping and Monitoring Program (FMMP), the United States Geological Survey (USGS) National Land Cover Dataset (NLCD) (1992), the California Department of Forestry and Fire Protection Fire and Resource Assessment Program Multisource Landcover Layers (MSLC), and the California Department of Water Resources (DWR) Land Use Survey. Importantly, CAML identifies 58 different crop types mapped by the California Department of Water Resources once or twice per decade in each county. Digital maps of these crop types are not available for historic conditions prior to 1990 except through back simulation (Harter et al., 2017). We selected the 1990 rather than the 2010 CAML map to account for some of the time difference between nitrate leached from landuse practices and the time of groundwater sampling (nitrate travel time) (Ransom et al., 2016). The nearly 60 agricultural and many non-agricultural landuse categories were aggregated into the following land type groups: Water & Natural, Citrus & Subtropical, Tree Fruit, Nuts, Cotton, Field crops, Forage Crops, Rice, Alfalfa & Pasture, Confined Animal Feeding Operation (CAFO), Vegetables & Berries, Peri-Urban, Grapes (including wine and table), and Urban. The Forage Crop group was further separated into fields likely receiving liquid manure irrigation and fields not likely to receive liquid manure based on proximity to CAFO landuse (within 1.6 km of dairy corrals, lagoons, or facility barns). This analysis does not take into consideration dry manure that may be exported off dairies and applied to crops. Our final study design had a total of 15 landuse and crop groups (hereby referred to as scenario 1). Alternatively, we also analyzed a scenario where CAFO landuse is grouped with (not distinguished from) Manured Forage Crops (14 landuse and crop groups, hereby referred to as scenario 2). The approach presented here can easily be modified to other landuse or management practice categorizations.

[revised manuscript text omitted]
 an estimate of recharge rates, groundwater concentrations are transformed to effective nitrogen loading rate distributions for each landuse group.

To determine whether the Bayesian model yields realistic estimates, we compare model results to two alternative, mutually independent datasets of nitrogen loading to groundwater: field measurements of nitrogen loading, obtained using a variety of field-based measurement techniques, and potential groundwater nitrogen loading obtained by closure to nitrogen mass balance based estimates of historic nitrogen fluxes in the CV. For further evaluation of the Bayesian loading estimates, we also consider hydrologic conditions other than mass loading that may affect nitrate concentrations measured in wells and used for the Bayesian loading estimation.

Studies determining nitrogen loading to groundwater from agricultural crops have historically used soil samples, anion exchange resin bags, suction lysimeters, or tile drain samples (Devitt et al., 1976; Embleton et al., 1979; Letey et al., 1977; Pratt et al., 1972; Pratt and Adriano, 1973; Adriano et al., 1972; Allaire-Leung et al., 2001; Liang et al., 2014). These field measurements are limited to a few crops and are not available for all crop groups analyzed in this study. Measurements were available for 5 crop groups: Citrus & Subtropical, Vegetables & Berries, Cotton, Alfalfa & Pasture, and Rice. Our Bayesian model results for these crop groups were generally consistent with the field measurements of nitrogen loading (Figure 7). For each crop group, multiple field measurements overlap with the 95% credibility interval (CI) of the Bayesian nitrogen loading estimates. For Rice, all three available field measurements overlap with the Bayesian loading model estimates. Overall, the field measurements encompass a wider range of values and extend to larger values than the 95% CI of our model estimates, especially for Vegetables & Berries.

The high variability of nitrogen loading rates measured (field measurements) within crop groups, especially for Citrus & Subtropical, Vegetables & Berries, and Cotton (Figure 7) is the result of several factors including within field crop rotation, variable irrigation and farming nutrient management practices within fields and among farms, and variable measurement methods. The field nitrogen loading measurements therefore need to be interpreted with some caution (Viers et al., 2012). The Bayesian loading estimates appear to confirm many of the field measurements, given the overlap of measured with estimated distribution of nitrogen loading. The range of loading rates predicted by the Bayesian model may therefore be interpreted as representing both, the potential variability of loading rates within a landuse group, and uncertainty about the loading rate.

Detailed spatially and temporally distributed nitrogen flux analysis for the Central Valley has been performed and documented for the Central Valley and Salinas Valley Groundwater Nitrogen Loading Model (GNLM) (Viers et al., 2012; Rosenstock et al., 2013; Harter et al., 2017). Briefly, the conceptual basis for the GNLM is a mass balance analysis of nitrogen fluxes into and out of agricultural crops, at the field scale, including nitrogen in atmospheric deposition, irrigation water, synthetic fertilizer, manure, wastewater effluent, harvest, runoff, and atmospheric emissions from soils. Potential groundwater nitrogen

loading from agricultural cropland was computed as closure to the mass balance. GNLM accounts for typical nitrogen fertilizer and harvest rates of 58 individual crops, spatially distributed across the CV. It also considers locally and regionally varying nitrogen deposition, irrigation water nitrate, and facility specific manure and waste water effluent applications to agricultural crops from CAFOs, wastewater treatment plants, and food processors. As a result, GNLM estimates of groundwater nitrogen loading within a crop group can be highly variable due to variability between crops within a group and due to local variability in non-fertilizer nitrogen fluxes. For consistency with the approach used here, we compare 1990 GNLM results to the Bayesian model results.

The Bayesian model results overlap with GNLM results for Citrus & Subtropical, Vegetables & Berries, Field Crops, Grapes, and the Water & Natural group (Figure 8). Median values for these crop groups obtained from the groundwater data (Bayesian analysis) were 39% (Citrus & Subtropicals), 20% (Vegetables & Berries), 10% (Field Crops), and 1% (Grapes) lower than mass balance based estimates (GNLM). In the Bayesian analysis, Citrus & Subtropical and Vegetable & Berries yielded the second, and third largest crop group median rates (scenario 1). The median GNLM rate is somewhat higher, but within the 68% CI estimated with the Bayesian model for both crop groups. The lower end of the range of GNLM estimates for these two crop groups is lower than predicted by the Bayesian model. The Bayesian model distribution extends to similar high concentrations as GNLM at the upper end of the predicted range for Citrus & Subtropical, but is about 50% lower than the upper end of the GNLM prediction for Vegetables & Berries (Figure 8).

The high loading rates estimated by our model for Citrus & Subtropical and for Vegetables & Berries appear to confirm the large difference between fertilizer and harvest rates for crops in this crop group. On the other hand, the similarity between our results and GNLM results for groundwater nitrogen loading from Field Crops and Grapes confirms the lower fertilization rates and resulting lower nitrogen surplus typically occurring in these latter crops (when not manured). For Citrus & Subtropical, the estimated high rates may also be a result of the significant potential for direct contamination pathways induced by farming practices thought to be common in the region where these crops are grown, along the eastern edge of the valley floor in Tulare and Fresno counties (Figure 6). This region includes a significant portion of "runoff" designated GWPA zones on soils that contain a shallow hardpan layer (Troiano et al., 2014) and where dry wells used for surface drainage are common (DeMartinis and Royce, 1990). In addition, the water table in these same regions is relatively shallow (7 – 10 m bgs) (DWR, 2011). Infiltration of agricultural surface runoff through dry wells and/or a shallow depth to water may lead to more rapid nitrogen loading at the high rates predicted by our model for this crop group than for other crops/landuses.

Bayesian model loading rates within the 95% CI of the Alfalfa & Pasture and Water & Natural groups had the lowest overall values (Figure 5 and Table 2). The Bayesian results for these crop groups are driven by the lack of apparent correlation between an increase in nitrate concentration in wells and increasing proportions of their respective area within well buffers (Figure 4). From a nitrogen mass balance perspective, low estimated nitrogen loading rates are expected for both landuse categories because fertilizers and manure are not typically applied to these areas: alfalfa is a legume, which has the ability to fix atmospheric nitrogen rather than relying on synthetic fertilizer; Water & Natural landuses are only subject to atmospheric nitrogen deposition and some symbiotic nitrogen fixation). In comparison, GNLM assigned a single value for groundwater

nitrogen loading from alfalfa (30 kg N ha$^{-1}$ yr$^{-1}$), based on reported field measurements (Viers et al., 2012). The assigned value in GNLM is much higher and outside the 95% CI estimated with the Bayesian model.

For urban landuse, GNLM assigned 20 kg N ha$^{-1}$ yr$^{-1}$ based on a review of urban nitrogen leaching (Viers et al., 2012), which is within our model estimated 95% CI for Urban nitrogen loading. In the Bayesian model results, Peri-Urban areas have a greater predicted 95% CI when compared to Urban (Figure 5). Peri-Urban areas are defined as rural homesteads. Each well buffer contained Peri-Urban areas. Peri-Urban areas were expected to have a greater nitrogen loading rate than Urban due to the use of septic systems. Septic systems are common outside urban areas not reached by centralized sewer services. Loading from these areas can be highly variable depending on septic system density. Our model estimated 95% CIs for Peri-Urban areas overlaps with the range for nitrogen loading from septic systems obtained by considering the density of households using septic, 10 to over 50 kg N ha$^{-1}$ yr$^{-1}$ (Viers et al., 2012). The Bayesian results for loading rates from Peri-Urban areas are consistent with research indicating that domestic wells in areas with higher septic system density are at significant risk to intercept septic system leachate (Bremer and Harter, 2012).

The CAFO landuse group, like the Citrus & Subtropical crop group, exhibited positive apparent correlation between nitrate concentration in wells and its area fraction in well buffers (Figure 4). In the Bayesian analysis (scenario 1), CAFO had the greatest estimated median loading rate among all crop and landuse groups (269 kg N ha$^{-1}$ yr$^{-1}$, Table 2). The value is about 50% higher than the value used for dairy corrals in the GNLM study (183 kg N ha$^{-1}$ yr$^{-1}$), and about one-quarter of the GNLM value for dairy lagoon loading to groundwater (1171 kg N ha$^{-1}$ yr$^{-1}$), which represents the average loading rate obtained from extensive field monitoring (Luhdorff and Scalmanini Consulting Engineers, 2015). However, both, dairy corrals and lagoons are included in the CAFO category for the Bayesian analysis. For the CV, Harter et al. (2017) estimated the total dairy corral area to be 12,200 ha and the total dairy lagoon area to be nearly 2,400 acres. Hence, the area weighted average loading rate from both areas is about 340 kg N ha$^{-1}$ yr$^{-1}$, well within the 68% CI of our Bayesian estimate. At 565 kg N ha$^{-1}$ yr$^{-1}$, the upper bound of the Bayesian 95% CI estimated for CAFO is much higher than average area-weighted corral and lagoon loading reflecting the large variability in groundwater nitrogen loading from this landuse apparent in groundwater nitrate values. Similarly, large variability has been observed in other research, particularly from dairy lagoons (Ham, 2002; Luhdorff and Scalmanini Consulting Engineers, 2015). VanderSchans et al. (2009) provided estimates specific to two dairies located on well-drained soil with shallow groundwater: 872 and 807 kg N ha$^{-1}$ yr$^{-1}$ for the dairy corral and dairy lagoons at the site, respectively. These estimates are greater than the 95% CI of the Bayesian model estimate, but confirm that the upper end of our estimated CI is not unreasonable.

Crop groups for which the groundwater nitrate based Bayesian model estimates are much lower than mass balance based GNLM estimates include Manured Forage, Nuts, Cotton, Tree Fruit, and Rice. GNLM results for these crops are driven mostly by the difference between applied synthetic fertilizer or manure and harvested nitrogen (Figure 8): For Rice, the Bayesian estimate is less than 5 kg N ha$^{-1}$ yr$^{-1}$, while GNLM predicts residual root zone losses to be 20 kg N ha$^{-1}$ yr$^{-1}$. Bayesian estimates of median loading rates for Cotton, Nuts, and Tree-Fruit range from 12 to 27 kg N ha$^{-1}$ yr$^{-1}$, while the GNLM estimates for these crop groups range between 90 and 110 kg N ha$^{-1}$ yr$^{-1}$. There is no overlap in CIs, between the two method estimates. However, measured field data for Cotton overlap with both method's CIs.

The discrepancy between the Bayesian analysis and other data for Rice, Tree Fruit, Nuts, Cotton, and possibly Manured Forage indicate that other processes, not explicitly accounted for in the Bayesian analysis, potentially attenuate the impacts of nitrogen mass loading, relative to field mass balance based estimates. In the Bayesian method, these processes, discussed below, lead to lower effective loading rate estimates when considering current groundwater quality data.

5      The distinct distributions obtained with scenario 1 simulations for CAFO and Manured Forage Crop indicates a statistically strong signal differentiating loading from these two landuse groups, despite the fact that the the two are typically located immediately adjacent to one another. CIs for CAFO and Manured Forage did not contain mutually overlapping values. The results correspond to differences found in previous research that estimated loading from manured forage crops to be nearly half of that from lagoons or corrals. VanderSchans et al. (2009, for the same location as the corrals and lagoons above) estimated

10   loading rates of 486 kg N ha$^{-1}$ yr$^{-1}$for Manured Forage crops. In a separate study, shallow groundwater monitoring well nitrate indicated leaching from manured forage fields of 280 kg N ha$^{-1}$ yr$^{-1}$ (Harter et al., 2002). While lower than CAFO estimates in these studies, both estimates are much larger than our Bayesian model estimates for Manured Forage crops in scenario 1, which estimates the upper end of the 95% CI to be less than 100 kg N ha$^{-1}$ yr$^{-1}$. GNLM estimates for manured crops also typically far exceed 200 kg N ha$^{-1}$ yr$^{-1}$. Our Manured Forage estimates may partly be lower due to an overestimation of land

15   area assumed here to be used for manure application (any forage field within 1.6 km of a dairy). Actual manure distribution in 1990 varied and may have taken up much less forage crop area. Non-manure forage may therefore be partially mixed into the category Manured Forage.

     Due to Manured Forage crops and CAFO typically being located adjacently, we also considered a scenario 2, where CAFO was lumped with Manured Forage into a single landuse category. The proportion of scenario 1 CAFO landuse within well

20   buffers was small (almost all wells had less than 10% CAFO landuse within the buffer, Figure 4), while Manured Forage crops occupied a larger proportion of the area (up to 50% of well buffers were Manured Forage crops, Figure 4). For CAFO landuse above about 0.05, higher nitrate concentrations were indicated, while higher concentrations were most dominant for Manured Forage landuse above 0.35 (Figure 4). Scenario 2 results represent an effective, area-weighted nitrogen loading across all dairy related landuses: corrals, lagoons, and manured crop areas. In the CV, Manured Crop areas are estimated to take up 174,000 ha,

25   more than a magnitude larger than the CAFO area (corrals and lagoons: 12,200 ha) (Harter et al., 2017). The much larger area of Manured Crops when compared to CAFO explains why the range of the estimated lumped nitrogen loading rate for Manured Crops and CAFO in scenario 2 is much closer to the range of scenario 1 Manured Forage Crops than to scenario 1 CAFO results. The process of merging CAFO landuse with the surrounding Manured Forage landuse reduces the estimated loading rates that would otherwise be specific to CAFO landuse (mostly lagoons and corrals), but may provide a more representative

30   estimate, given the uncertainty about past manure application areas, for CAFO and associated (partially) manured areas as a whole. We note that the similarity of results between scenario 1 and scenario 2 obtained for other crop and landuse groups indicates that the Bayesian method is robust to the particular choice of crop and landuse groupings.

     The discrepancy between some mass balance estimates and the Bayesian model estimates may be due to several hydrologic processes that affect well nitrate concentrations independent of nitrogen loading rates in the recharge area of a well.

35   These include dilution with older groundwater, mixing with recharge water from streams, and denitrification or ammonium

volatilization in the vadose zone or in groundwater (Ransom et al., 2017). Dilution of nitrogen in recharge water is most likely to occur through mixing along the well screen with older, low nitrogen containing, water. Mixing with old water (that recharged prior to the advent of nitrogen fertilizers in the 1930s and 1940s) within well screens could potentially have affected the model estimated loading rates for all crop and landuse groups. Due to the length of well screens, all domestic well samples contain
5  water of mixed age. A study located near Fresno, CA (within our study area) found that groundwater samples from individual wells contained groundwater with an age range typically greater than 50 years (Weissmann et al., 2002). Weissmann et al. (2002) attributed the high variance in groundwater residence time within a single well to the heterogeneity within the alluvial aquifer system which produced spatially varying flow velocities. Weissmann et al. (2002) also reported significant positive skewness (tailing) in the distribution of groundwater ages within individual wells, meaning wells contained some groundwater
10  which was much older than the median age. The authors reported the tailing behavior was due to low hydraulic conductivity units within the aquifer in which slow advection and diffusion dominate the transport process. These results are similar to an earlier study in the Salinas Valley, California (an alluvial aquifer system that, at comparable depth, is similar to the CV and dominated by agriculture) where the authors found significant dispersion of groundwater ages within simulated groundwater samples (Fogg et al., 1999). Simulated water samples from Fogg et al. (1999) had groundwater ages ranging from 10 years to
15  greater than 500 years. The authors point out that the water pumped from wells in the Salinas Valley was only partially from water that was young enough to be contaminated by nitrate and that this proportion would only increase in the future.

Geostatistical analysis of groundwater age tracers from wells sampled in the CV has estimated the depth to the top of well screens pumping pre-modern (age of 60 years of more) groundwater to be between 30 - 120 m (Visser et al., 2016). According to those results and considering the median depth to bottom of well screen for wells in our study (64 m), a portion of our study
20  wells screened intervals likely penetrate the interface between young and old water. Therefore, mixing within wells with water recharged prior to the intensive use of fertilizers, with water with long residence times (tailing effect), or with water recharged between the 1940s and 1970s, when nitrogen losses for many crops were smaller than in 1990 or later (Harter et al., 2017) could have led to the lower estimates of nitrogen loading for some crops in this study. Mixing with groundwater recharged prior to 1990 may play a significant role in the Bayesian estimates obtained for Manure Forage, Cotton, and Nuts: Significant increases
25  in manure application to forage crops occurred only after the 1960s, with large increases in the 1980s and 1990s (Harter et al., 2017). These changes may not yet have affected much of the water drawn from the measured wells. For Cotton and Nuts, field mass balance based estimates for nitrogen loading indicate much lower median rates in 1975 and 1960 (about 40 kg N ha$^{-1}$ yr$^{-1}$). Also, the harvested area for nut crops increased sharply between 1960 (64,000 ha) and 1990 (250,000 ha) (Harter et al., 2017). At individual wells, or even across our set of wells, it is difficult to further assess the dilution effect with older water
30  without more detailed analysis of groundwater age throughout the CV and more information on study well screened intervals.

Infiltrating river water with very low nitrate concentration is a significant source of recharge in some areas. This may also dilute otherwise high nitrogen concentrations in land surface recharge. Wells near rivers may receive a significant fraction of river recharge. A study focused on the TLB (within the CV) geospatially related areas near rivers with lower nitrate concentration in wells. The study highlighted areas where major rivers flow into the TLB from the Sierra Nevada Mountains and found
35  that these areas were also characterized by wells with lower relative nitrate concentrations (Boyle et al., 2012). A statistical

analysis of CV groundwater nitrate recently confirmed that proximity to major streams is a significant controlling factor for a wells' nitrate concentration (Ransom et al., 2017).

Boyle et al. (2012) point out that agricultural areas near rivers are also more likely to receive surface water irrigations. Nitrogen loading from fields receiving low nitrate surface water irrigations is likely to be lower than from fields irrigated with nitrate contaminated groundwater (Boyle et al., 2012). Irrigation water source may have impacted our model estimates and resulted in the lower loading rates compared to mass balance estimates for some crops.

Denitrification and ammonium volatilization could also play an important role in some differences between our model estimates and mass balance estimates, though we do not suspect widespread regional denitrification. A study focused in the San Joaquin Valley correlated anoxic groundwater conditions to lower nitrate concentrations, but the authors did not attribute this to denitrification (Landon et al., 2011). Instead, Landon et al. (2011) attributed the lower nitrate concentrations in wells with water classified as anoxic to older groundwater with longer residence times (recharged prior to the intensive use of fertilizers). Landon et al. (2011) did not find significant decreases in nitrate concentration in wells due to denitrification. Results of a multi-model averaging approach to estimate oxygen and nitrogen reduction rates in the San Joaquin Valley did estimate denitrification rates to be significant (Green et al., 2016). However, Green et al. (2016) also estimated oxygen reduction rates to be low with a median of 0.12 mg $L^{-1}$ $yr^{-1}$. Much of the shallow groundwater in the CV is well-oxygenated: the dissolved oxygen content of Lockhart et al. (2013) study wells with a measurement (Table 1) was above 5 mg/L on average (Ransom et al., 2016). In addition, Green et al. (2016) found that the estimated rates of oxygen and nitrogen reduction would not protect wells from nitrate contamination, given current nitrogen application rates. We therefore do not expect that denitrification or ammonium volatilization had a significant, overall effect on our model results, but rather may have had an isolated effect in areas with Rice and possibly with Manured Forage. For example, a study on four rice fields in the Sacramento Valley (northern CV) found little to no nitrate leaching below the rice root zone (pore water nitrate levels were typically below approximately 2.5 mg/L $NO_3$-N). This was attributed to denitrification during the rice growing season when fields are flooded, ammonia volatilization, plant uptake, and crop management practices that contribute to the development of a hardpan layer directly below the rice root zone. The study also found very low nitrate concentrations in groundwater wells near rice fields (median value less than 1 mg/L $NO_3$-N) (Liang et al., 2014) consistent with our estimates of Rice nitrogen loading. The GNLM mass balance estimates are outside the range of our model predicted CIs for Rice as they reflect nitrogen losses prior to denitrification or ammonium volatilization potentially taking place in saturated clay soils of rice fields.

Denitrification may also explain why the attenuation factor for areas outside GWPA protection areas is significantly lower than 1 (Figure 2). Regions outside GWPAs are characterized by larger depth to groundwater (greater than 21 m). The 15% to 30% lower apparent nitrogen concentrations in the less vulnerable regions may be due to longer travel times in the deep vadose zone or some additional denitrification and ammonium volatilization in the heavier soils or the underlying deep vadose zone, or in groundwater.

Our model estimates a greater median groundwater recharge rate ($r$) compared to the prior information from the CVHM model. This is likely because the study wells are concentrated in agricultural areas with greater recharge rates due to irrigation. The CVHM (Faunt, 2009) estimated recharge rates are calculated for the entire Central Valley, including natural areas with few

wells and little agriculture. Many of our study wells were spatially clustered in the Tulare/Kern and Kings subbasins, which had median CVHM estimated recharge rates of 0.21 and 0.35 (m year$^{-1}$), respectively for the 1990 decade. These median rates are near our model estimated median recharge rate of 0.281 (m year$^{-1}$) (Figure 3).

**6 Conclusions**

5   The novel Bayesian tool developed here provides a robust statistical methodology to relate nitrate measurements in wells to the various types of surrounding landuses as a means to obtain a statistical distribution of nitrate loading rates. After accounting for some hydrologic processes not explicitly represented in the approach (denitrification, ammonium volatilization, mixing with older water or water recharge from streams) the Bayesian model estimates were consistent with previous independent estimates and measurements of potential groundwater nitrogen loading. The validation against independently obtained data demonstrate

10  the general usefulness and accuracy of the Bayesian nonpoint source pollutant loading model introduced here. The information can provide a better assessment of landuse impacts to water quality based on extensive nitrate and other nonpoint source groundwater contaminant data measured in private wells. The tool can be used to define high nitrogen loading (high risk) zones (Figure 6). As is apparent from Figures 1 and 6, much of the CV already suffers from or is at risk for serious groundwater contamination by nitrate. Our results indicate that the highest nitrogen loading rates are associated with Confined Animal

15  Feeding Operations (dairies) and their associated feed crops (with the exception of alfalfa), as well as from Citrus & Subtropical crops and Vegetable & Berry crops. Yet, interactions between the depth to older water, well construction, direct contamination pathways, groundwater depth, presence of river water recharge and landuse have likely affected the amount of nitrate pumped by wells. Estimates of nitrate loading generally correspond to previous field measurements or mass balance estimates. For Nuts, Cotton, Tree Fruit, and Rice estimated nitrogen loading rates were lower than mass balance estimates. Nuts, Cotton, and Tree

20  Fruit estimates may have been affected by dilution of crop leachate water past the root zone by infiltrating low-nitrate river water or by mixing with older low-nitrate water within the well screen. Land managers may default to the mass balance estimates for those crops. Rice estimates were likely lower than mass balance estimates due to denitrification and ammonium volatilization directly below saturated rice fields, which mass balance estimates did not consider. Estimates of nitrate leaching concentration for particular crop and landuse types, obtained with this tool may not be generalized and transferred to regions with substantially

25  different climate, agronomic, geologic, geomorphic, or soils conditions. However, the statistical modeling approach provided here is broadly applicable to other semi-arid, irrigated regions underlain by alluvial aquifers and to nonpoint source pollutants other than nitrate, e.g., salinity and pesticides. Our results could potentially be improved with more information on groundwater age and portion of older water pumped by the wells in our study. Also, a potential limitation of the method is the limited availability of historic crop and landuse maps of sufficient resolution, corresponding to typical groundwater age found in wells.

30  In a similar vein, regional scale models capable of simulating physical groundwater flow processes described by Weissmann et al. (2002) and Fogg et al. (1999) are needed to more accurately access the transport of nitrogen in the subsurface.

**7 Code availability**

Model code may be available upon request.

**8 Data availability**

This dataset is available upon request. Please note well location information, well data group, or well names are not publicly
5    available due to confidentiality agreements with well owners.

*Author contributions.* Dr. Thomas Harter designed the research question and experimental, conceptual, and overall statistical structure. Katherine M. Ransom, with assistance from Quinn Barber, prepared the initial well data. Andrew M. Bell processed landuse layers for the dataset. Katherine M. Ransom with assistance from Mark N. Grote and Arash Massoudieh wrote and ran model code and assessed model results. George Kourakos and Thomas Harter prepared and processed Groundwater Nitrogen Loading Model (GNLM) results for use in this
10    study. Katherine M. Ransom prepared the manuscript text and figures.

*Competing interests.* The authors declare that they have no conflict of interest

*Acknowledgements.* We are extremely grateful to Mark N. Grote and Arash Massoudieh for their help with model development and assessment, our work would not have been possible without you. Thank you to Jo Ann M. Gronberg for preparing figure map templates of the Central Valley, California and to Claudia C. Faunt and Jon Traum for providing and processing the CVHM data for our initial estimates
15    of groundwater recharge rates. Partial funding for this work was provided through the State Water Resources Control Board (SWRCB) Agreement No. 09-122-250, and through a grant from the California Department of Agriculture Fertilizer Research and Education Program, project numbers 11-0301 and 15-0454. The authors gratefully acknowledge the financial support of the Center for Watershed Sciences made possible by a gift from the S. D. Bechtel, Jr. Foundation.

[Figure]

**Figure 1.** Study well locations color coded by nitrate (NO$_3$-N mg/L) value overlain with CDPR GWPA zones (runoff and leaching undifferentiated).

[Figure]

**Figure 2.** Posterior probability density for the non-GWPA attenuation factor, $k$, for scenario 1 (tan) and 2 (teal).

[Figure]

**Figure 3.** CVHM estimated annual vertical groundwater recharge (grey bars), log-normal prior probability density for recharge input to model (grey line) and model estimated posterior probability density for the annual recharge rate for scenario 1 (tan) and 2 (teal).

[Figure]

**Figure 4.** Scatterplot of proportion of landuse within each well buffer versus well nitrate concentration for each of the 15 landuse or crop groups in scenario 1. The red line is a locally weighted scatterplot smoothing line (Cleveland, 1979). Note that each plot shows nitrate concentrations between 0 and 25 mg/L NO$_3$-N (approximately 5 times the median value), however all data was used to calculate the plotted smoothing lines. The x-axis is scaled differently among subplots for better resolution.

[Figure]

**Figure 5.** Posterior probability densities of estimated nitrogen loading for scenario 1 (tan) and 2 (teal). 95% Credibility intervals are represented by the light grey shading (dark grey shading occurs where scenario 1 and 2 estimates overlap).

[Figure]

**Figure 6.** CAML landuse for the 15 landuse groups in scenario 1 (left side) and the same landuse groups keyed to the median estimated nitrogen loading rate in kg N ha$^{-1}$ yr$^{-1}$ for the corresponding group (right side).

[Figure]

**Figure 7.** Credibility intervals of posterior probability densities of estimated nitrogen loading rates for selected crop groups plotted with historical direct measurements of nitrogen loading from California ((Devitt et al., 1976; Embleton et al., 1979; Letey et al., 1977; Pratt et al., 1972; Pratt and Adriano, 1973; Adriano et al., 1972; Allaire-Leung et al., 2001; Liang et al., 2014). Thinner lines are 95% credibility intervals, thicker lines are 68% credibility intervals, the solid dot is the median estimated value, and the open black circles are historical field measurements.

[Figure]

**Figure 8.** Credibility intervals of posterior probability densities of estimated nitrogen loading rates for selected crop groups plotted with the results from the Groundwater Nitrogen Loading Model (GNLM) (Viers et al., 2012; Rosenstock et al., 2013) (black). Thinner lines are 95% credibility intervals, thicker lines are 68% credibility intervals, and the dot is the median estimated value.

**Table 1.** Original data source, number of wells, and well location method for private wells included in final database (2149 wells total).

| Dataset Group | Dataset Subgroup | Number of Wells | Location Method/Accuracy |
|---|---|---|---|
| CASTINGS, CVRWQCB Fresno Office | Private wells on dairies | 361 | Located at reported coordinates of the dairy, the reported street address of the dairy, or the centroid of dairy parcel(s) (single, multiple adjacent parcels, or centroid of multiple non-adjacent parcels) (Boyle et al., 2012). |
| CASTINGS | GAMA Domestic Tulare County | 134 | Well locations randomly offset by 1/2 mile from true location (Boyle et al., 2012). |
| CASTINGS | Department of Pesticide Regulations (DPR) | 62 | Located at the centroid of the United States Public Land Survey System (PLSS) section (approximately 1 mi$^2$) in which the well resides (within 1/2 mile of the actual well location) (Boyle et al., 2012). |
| CASTINGS | Fresno County | 295 | Located at street address reported on well logs or centroid of the reported Assessor's Parcel Number (APN) (Boyle et al., 2012). |
| CASTINGS | The U.S. Geological Survey's (USGS) National Water Information System (NWIS) | 17 | Unknown (Boyle et al., 2012). |
| CASTINGS | Tulare County Environmental Health | 437 | Located at centroid of the reported APN (Boyle et al., 2012). |
| Lockhart et al. (2013) | None | 200 | Geographic coordinates digitized with imagery from Google Earth (Lockhart et al., 2013). |
| GAMA Domestic | Tehama, El Dorado, and Yuba Counties | 253 | Well locations randomly offset by 1/2 mile from true location. |
| CVRWQCB Rancho Cordova Office | Private wells on dairies | 390 | Geocoded using street address. |

**Table 2.** Median and 95% credibility interval bounds for estimated nitrogen loading rates by group for scenario 1 and 2.

| Scenario 1 Group | Median | Lower Bound | Upper Bound | Scenario 2 Group | Median | Lower Bound | Upper Bound |
|---|---|---|---|---|---|---|---|
| Water & Natural | 4.0 | 2.1 | 7.4 | Water & Natural | 4.3 | 2.2 | 7.5 |
| Citrus & Subtropical | 64.5 | 35.7 | 113.8 | Citrus & Subtropical | 69.0 | 37.0 | 116.5 |
| Tree Fruit | 12.2 | 4.6 | 26.2 | Tree Fruit | 13.0 | 4.7 | 26.2 |
| Nuts | 24.8 | 12.6 | 46.9 | Nuts | 26.7 | 13.8 | 46.8 |
| Cotton | 12.7 | 4.6 | 28.1 | Cotton | 12.7 | 5.3 | 25.8 |
| Field Crops | 31.6 | 10.7 | 70.1 | Field Crops | 28.8 | 9.4 | 71.4 |
| Non-manured Forage | 23.3 | 7.7 | 54.3 | Non-manured Forage | 18.1 | 4.4 | 44.9 |
| Manured Forage | 45.6 | 20.4 | 89.3 | Manured Forage & CAFO | 83.4 | 43.2 | 146.6 |
| Rice | 4.1 | 0.9 | 12.7 | Rice | 4.2 | 0.9 | 11.8 |
| Alfalfa & Pasture | 3.7 | 0.2 | 13.2 | Alfalfa & Pasture | 5.0 | 0.3 | 13.5 |
| CAFO | 268.9 | 118.5 | 565.1 | NA | NA | NA | NA |
| Vegetables & Berries | 49.1 | 18.7 | 102.8 | Vegetables & Berries | 61.4 | 25.7 | 122.5 |
| Peri-Urban Areas | 26.0 | 8.8 | 59.2 | Peri-Urban Areas | 30.1 | 12.3 | 68.6 |
| Grapes | 24.1 | 12.6 | 44.1 | Grapes | 27.1 | 14.0 | 49.3 |
| Urban | 13.9 | 6.6 | 26.8 | Urban | 14.2 | 6.5 | 27.9 |

NA: Not applicable.